

# A close look at using national ground stations for the statistical modeling of NO$_2$

Foeke Boersma and Meng Lu

Department of Geography, University of Bayreuth, Universitätsstraße 30, 95447 Bayreuth, Germany

**Correspondence:** Foeke Boersma (foekeboersma@hotmail.com)

**Abstract.**

Air pollution causes a manifold of negative health and societal problems. It is therefore essential to model and predict air pollution over space. An increasing number of statistical models of air pollution have been developed using geospatial variables associated with air pollution emission and dispersion processes. However, the increasing number of air pollution models does not always equate to an increase in prediction accuracy and uncertainty reduction. An important aspect that is often disregarded is the spatial heterogeneity. In this study, we aim to evaluate and compare various spatial and non-spatial statistical and machine learning methods, with attention given to different spatial groups. Spatial groups are identified by the predictor variables. We found that prediction accuracy differs substantially in different spatial groups. Predictions in places close to roads with high populations show poor prediction accuracy, while prediction accuracy increases in low population density areas for both local and global models. Prediction accuracy is further increased in places that are far from roads for global models. This division into spatial groups also shows that global non-linear methods are capable of higher prediction accuracy than global linear methods. The spatial prediction patterns show that non-linear methods generally predict more smoothly than linear methods. Additionally, clusters of predicted air pollution differ within and between cities. Lastly, applying the same methods to the local dataset yields poor metrics, especially for the non-linear methods.

## 1 Introduction

The importance of air quality assessment cannot be over-stressed as air pollution causes a manifold of negative health effects to people such as increasing the risk of heart diseases, lung cancer, strokes, acute and chronic respiratory diseases (Núñez-Alonso et al., 2019) and is a leading mortality cause (Ritchie and Roser, 2017). An estimated 5 million deaths are caused by air pollution in 2017, meaning that nearly 1-in-10 deaths are due to air pollution (Ritchie and Roser, 2017). Additionally, He et al. (2022) state that about 55,000 and 417,000 premature deaths in 41 European countries in 2018 are related to NO$_2$ and PM2.5 respectively. Recently, several studies argue that a link between air pollution and the pandemic COVID-19 exists. According to Medical News Today (2020), deaths related to COVID-19 cases spiked after significant increases in air pollution in cities such as London and Paris. Also, World Bank (2020) concludes that people in polluted areas are twice as susceptible to COVID-19 infections. Next to the negative health effects, architectures in urban area encounter negative effects that are caused by air pollution. Increasing air pollution concentrations augments damage to building materials and cultural objects (Núñez-Alonso



et al., 2019). The negative effects of air pollution have also been analyzed from an economic perspective (Dechezleprêtre et al., 2019). In a study focusing on European countries, it was found that "an 1g/m3 increase in PM2.5 concentration causes a 0.8% reduction in real GDP that same year" (Dechezleprêtre et al., 2019, p. 3). The relationship between increasing air pollution and decreasing economic activity is mainly because of a reduction in output per worker, which is related to reduced labour
productivity or greater absenteeism due to the influence of air pollution (Dechezleprêtre et al., 2019).

Quantifying air quality forms the base for environmental management and the understanding of the negative effects of air pollution. Air pollutants have been modelled at different spatial scales, up to globally. The models can be classified into statistical models, chemical transportation models and air dispersion models. The chemical transportation models are developed for large-scale air pollution modeling. The air dispersion models require detailed and spatially revolved emission data to model
small-scale spatial variations in air pollutants sufficiently (Beelen et al., 2013). Statistical modeling is becoming more popular in high-resolution mapping at different spatial scales, due to an increment in available predictors (known as GIS variables) and computational capability. Land Use Regression (LUR) is the most well-known statistical modeling approach for air pollution. It is based on a linear regression model to capture the spatial variability of traffic-related air pollution in urban areas whereby most of the models are based on measurements from ground monitoring stations (Hoek et al., 2008; Wang et al., 2020).
Geostatistical methods such as kriging could further capture the spatial correlations between the observations. However, a number of studies favor the simple character of the LUR models and conclude that it performs better than or equivalent to geostatistical methods (Hoek et al., 2008; Marshall et al., 2008; Beelen et al., 2013). These counter-intuitive results may be due to that the geostatistical models are not optimally specified.

Although the linear models have the advantage of being highly interpretable and can be extrapolated, they may not capture
the complex air emission, dispersion, and deposition process (Wang et al., 2020). Data-driven, nonparametric models (most commonly known as machine learning methods in air pollution mapping), such as tree-based algorithms have been applied to air pollution mapping in recent years, as these methods may better capture the non-linear relationships between pollutants and predictors (Weichenthal et al., 2016; Reid et al., 2015; Lu et al., 2020). Brokamp et al. (2017) compare the Land Use Random Forest models (LURF) with the LUR models for elemental components of PM 2.5 in the urban area of Cincinnati, Ohio,
and find that the LURF shows a lower prediction error variance for each elemental model with cross-validation. Kerckhoffs et al. (2019) report that machine learning algorithms such as bagging and random forest, explain more variability in ultra fine particle concentrations than multiple linear regression and regularized regression techniques. Ameer et al. (2019) advocate that the random forest regression is the best technique, compared to decision tree regression, Multi-Layer Perceptron regression, and gradient boosting regression, for pollution prediction for data sets of varying size, location, and characteristics. Ren et al.
(2020) conclude that non-linear machine learning methods achieve higher accuracy than the linear LUR, thereby stressing that a careful design of hyperparameter tuning and flexible data splitting and validations is important to acquire stable and reliable results. Chen et al. (2019) compare 16 algorithms to predict the annual average fine particle (PM2.5) and nitrogen dioxide ($NO_2$) concentrations across Europe, and also conclude with a favor of the ensemble tree-based methods. However, this difference is more prevalent for the PM2.5 pollutant while $NO_2$ model predictions show a similar $R^2$. At the same time,
a high correlation is reported between the predicted values of the various models used in the study. Furthermore, as they



measure two pollutants, the most influential predictors on the variation of the pollutant, differ substantially. To extend, satellite observations and dispersion model estimates are among the highest influential predictors for PM2.5 concentrations while the variety in $NO_2$ is for a substantial part caused by traffic-related variables. The major contribution of road traffic to $NO_2$ concentrations is supported by Wong et al. (2021). Additionally, they mention that the rise of $NO_2$ might be related to a specific

type of diesel particulate produced by heavy vehicles such as buses, whereas nitrogen is produced by, particularly long-range transport, passenger cars that are gasoline fuelled. Consequently, they opt that local $NO_2$ concentration levels can be improved by reducing road traffic.

Recent years have seen a considerable increment in the number of studies applying statistical modeling to air pollution mapping and a blossoming of local and global air pollution maps, as well as the application of these maps in urban and

health studies. However, the evaluation of the air pollution models and maps remains a challenge. One reason is the lack of air pollution measurements. A second reason may be attributable to a different focus in the domain of spatial heterogeneity in air pollution. For instance, He et al. (2022) acknowledge spatial heterogeneity in measurement stations as they show that the probability density function of concentrations (NO, $NO_2$, PM10, PM2.5) of different spatial categories (urban traffic; suburban/rural traffic; urban industrial; suburban/rural industrial; urban background; suburban background; rural background)

show different patterns. However, the research does not focus on modeling thus potential differences in prediction accuracy for every concentration type and spatial category. Another reason why evaluating air pollution models and maps remains a challenge, is that most of the current statistical modeling approach only assessed the overall accuracy, but not the accuracy over space (Hoek et al., 2008; Chen et al., 2019). Hoek et al. (2008) state that a LUR model typically explains 60-70% of the variation in $NO_2$. However, the amount of explained variation may be very low in areas close to traffic. Chen et al. (2019) argue

that most of the previous air pollution exposure assessment studies make no distinction in the characteristics of the monitoring sites when performing cross-validation, potentially leading to misrepresenting model results. Therefore, they opt to "evaluate models using pollution data collected from monitoring sites which represent the application locations" (Chen et al., 2019, p.3). Shaddick et al. (2020) also argue that the uncertainty in air pollutant measurements is only discussed in a couple of studies. A consequence of the inadequate evaluation is that the non-extrapolating property of most non-parametric machine learning

methods is commonly ignored. Areas to be predicted could differ considerably in their societal and environmental properties compared to the training data, yielding highly biased predictions, which are not evaluated in multiple studies (Shaddick et al., 2020).

Given the increasing number of modeling and prediction techniques, and the presence of misrepresented prediction maps due to heterogeneity issues, this study aims to understand *to what extent can statistical models be used in predicting $NO_2$*

*concentrations given high-quality, high-temporal resolution ground station measurements: how does the performance of statistical models differ and how does it differs spatially?* The study area is in the Netherlands and Germany. Two datasets are used. One is the official national ground station measurements of the two countries (EEA, 2021), and the other is the ground station measurements of the Amsterdam area (Gemeente Amsterdam, 2022). The aims are to compare modeling accuracy and prediction patterns 1) using the national ground stations and a set of more densely distributed local ground stations; 2) in areas





close and far away to traffic and with different population densities, and 3) to using different statistical models considering, and understand the added value of modeling spatial correlation.

## 2  Methodology

### 2.1  Data

We refer to the dataset which consists of all the ground station measurements in Germany and the Netherlands as the "global
dataset", and the dataset which consists of ground station measurements in the Amsterdam area as a "local dataset".

National air quality ground monitoring stations in the Netherlands and Germany are used in this study which is also apparent in the study of Lu et al. (2020). Supplementary, figure 1a shows the equal divided area that is covered for each sample. The minimal area is 1,11 km$^2$; the maximum area is 4118,5 km$^2$ and the mean is 811,53 km$^2$. Interestingly, the urban areas give lower values meaning that more measurement stations are situated there. To evaluate the global model and examine the
differences between the global- and local model, the urban area of Amsterdam is used. Additionally, the direct, less densely populated areas around Amsterdam are included to examine the effect of the urban area on the predicted NO$_2$ concentration levels per local model.

In order to test whether the prediction quality differs between areas with different spatial characteristics (e.g. high vs. low
road density). The observations of the global and local datasets are split into three spatial groups. The three groups are defined as follows:

1. Urban: observations that belong to this group are within 100 meters of either road class 1 (highway) or 2 (primary roads) and the population 1000 (population density within 1000 meter of every measurement station) values are in the highest 25%; or the road class 3 (local roads) values are in the highest 25%; and the population 1000 values are in the highest
115   25%.

2. Low population: observations that belong to this group are within 100 meters of either road class 1 or 2 and the population 1000 values are in the lowest 75%; or the road class 3 values are in the highest 25% and the population 1000 values are in the lowest 75%.

3. Far from roads: observations that belong to this group are further than 100 meters of either road class 1 or 2; or the road
class 3 values are in the lowest 75%.

By applying this division [1], 85 observations were classified as "urban", 138 as "low population" and 259 as "rural", together compromising the 482 observations of the global dataset. Since the local dataset contains fewer samples and applies more to the urban area (of Amsterdam), the threshold is adjusted from 0.75 to 0.5 (i.e. "urban" is now related to the 50% highest

---

[1]The related code is visible in supplementary, Code 1





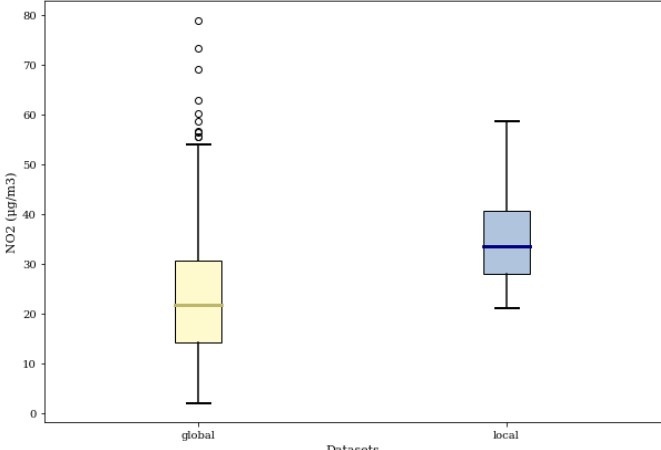

**Figure 1.** NO$_2$ distributions of global- (yellow) and local (blue) model

values rather than the 75% highest values). For the local dataset, 56 observations are attributable to the spatial group "ur-
ban", 46 to "low population", and 30 to "far from road". Supplementary, figure 2 and figure 3 display the spatial distribution of
observations through spatial groups, both for the global- and local datasets, thereby including information on the spatial groups.

*Air pollution ground station measurements*

The mean of the NO$_2$ for the year 2017 in the global dataset is 23.70; with a standard deviation of 12.89. The minimum
is 2.12 and the maximum is 78.88. For the local dataset, the mean (35.23) is considerably higher but the variance is less
pronounced (8.90). The minimum is 21.00 and the maximum is 58.50. The data distribution for both datasets, in terms of
NO$_2$, is displayed in figure 1. The related spatial distributions of NO$_2$ values for the global and local datasets are displayed in
supplementary figure 4 and figure 5 respectively.

*Spatial predictors*

Table 1 shows the obtained data that are converted to predictors that potentially influence NO$_2$ concentration levels. A set
of variables with related data is already derived from Lu et al. (2020), including industrial areas from OpenStreetMaps, road
length from OpenStreetMaps, population from GHS-POP R2019A population grid, and Earth night light from VIIRS in vari-
ous buffers, wind speed and temperature at 2 m altitude from ERA-LAND 5 climate re-analysis model, elevation from 30 m
Radar global product, Tropomi level 3 NO$_2$ of 2018 and global radiation. An overview of the variables derived from Lu et al.
(2020) can be found in supplementary, table 1.

We used the precipitation from weather stations (National Centers for Environmental Information, 2017) and conducted spatial



interpolation using ordinary Kriging to cover the NO$_2$ measurement stations. The precipitation consists of average monthly
precipitation data, measured in millimeters.

The "World Settlement Layer 2015" is used in order to determine the building density. This layer is publicly available on
figshare (Marconcini et al., 2020). Taking building density as an explanatory variable, multiple studies make use of several
measurement scales, consisting of buffers that vary in size. Beelen et al. (2013) quantify several buffers that represent building
density whereby parameters are 100m, 300m, 500m, 1000m and 5000m. Another study performed by Kheirbek et al. (2014)
measured the correlation between air pollution and building density via 15 circular buffers, ranging from 50m to 1000m. Lu
et al. (2020) also make use of buffers for one explanatory factor in order to encourage comprehensiveness within the method-
ology. Varying buffer sizes are implemented in different studies and also in our study. As several measuring stations are within
the vicinity of each other, especially in urban areas, the maximum buffer size is set to 1000m. Moreover, two other circular
buffers have parameters of 100m and 500m.

The values for the Normalized Difference Vegetation Index (NDVI) are obtained through NASA and are related to 2017.

The Dutch dataset for traffic volume is obtained via Nationaal Dataportaal Wegverkeer (NDW) whereas the German dataset
for traffic volume is obtained via Bundesanstalt für Strassenwesen (BAST). Both the NDW- and BAST-datasets are generated
via automatic counting stations. Both datasets are represented as a point-feature shapefile. The traffic volume is expressed in
average hourly traffic. The traffic volume values of the traffic counting stations are assigned to roads, which are represented as
lines, consisting of all highways and major roads in the Netherlands and Germany. The traffic volume values are calculated per
buffer, ranging from 25m, 50m, 100m, 400m, and 800m. The average hourly traffic $T$ per buffer is calculated as:

$$T_x = \frac{\sum(l_{ix} * T_{ix})}{\sum(l_{ix})} \tag{1}$$

Whereby $l$ corresponds to the length of road segment $i$ in buffer $x$.

## 2.2 modeling NO$_2$ globally and locally

For the global models, we first selected features based on the SHAP method (Shapley, 1953) and by means of a 10-fold cross-
validation. The variable selection aims at removing irrelevant or colinear predictors which would otherwise generate unstable
estimates (Araki et al., 2018). The average median for each feature is ranked. Once the ranking of each feature is determined,
the number of features that composes the model is examined by evaluating the Root Mean Square Error (RMSE), R$^2$, and Mean
Absolute Error (MAE) for a different number of predictors using the random forest algorithm, the iteration starts with the two
most influential features and ending with the thirty most influential features, with an iteration step of 1. The global models (i.e.
using the annual average of the NO$_2$ measurements from the 482 ground stations across The Netherlands and Germany) are
compared between the random forest model, the lightgbm model and the xgboost model, representing tree-based methods, and
the lasso and ridge-models, representing linear models. The local dataset consists of 132 measurement stations in or around



| Variable | Data | Location | Time | Source |
|---|---|---|---|---|
| Precipitation | Global Summary of the Month (GSOM), Version 1 | The Netherlands, Germany | 2017 | National Centers for environmental information |
| Vegetation | MODIS/Terra Vegetation Indices Monthly L3 Global 1km SIN Grid V006 – MOD13A3 | The Netherlands, Germany | 2017 | Earth data - NASA |
| Building density | building density - World settlement footprint | The Netherlands, Germany | 2015 | Marconcini et al., 2020 |
| | Automatic counting stations | Germany | 2017 | Bundesanstalt für Strassenwesen |
| Traffic volume | Traffic counting | The Netherlands | 2017 | Nationaal Dataportaal Wegverkeer *Note: account is needed since data does not originate from NDW-open dataportal* |

**Table 1.** Data - descriptives

Amsterdam. In an attempt to make better use of the spatial information, we added the universal kriging method and mixed-effects modeling. At the same time, we implemented the ordinary kriging method for comparison. The kriging parameters are

fit using the R package automap (Hiemstra et al., 2008). Details are found in supplementary, figure 6 and figure 7. For both the universal kriging and linear modeling method, a model is made that does not account for the spatial groups and a model is made that does take the spatial groups into account. When accounting for spatial groups, a model is created for each spatial group from which $NO_2$ will be derived. For the mixed effect model, the variable that describes the spatial group to which every observation belongs is used as the random effect. Eventually, eleven models are used and compared, five of which are based

on the global dataset; six of which derive from the local dataset. Furthermore, the prediction patterns between the local- and global mappings are compared. Hereby, the mobile $NO_2$ map is used as benchmark to evaluate the accuracy of every model (Kerckhoffs et al., 2019).

### 2.3  Statistical models

#### 2.3.1  Ensemble trees

The original dataset of Lu et al. (2020), complemented with additional variables (table 1) serves as input for the random forest, lightgbm, and xgboost algorithms. In order to create a prediction model for $NO_2$, the algorithms of random forest, lightgbm, and xgboost are used and compared with each other. In the study of Lu et al. (2020), the random forest tree techniques perform





better in predicting the non-linear relationship between $NO_2$ and its predictors than linear regression models. As Lu et al. (2020) argue: "In general, the tree-based methods obtained higher accuracy compared to the regression-based methods, especially the rIQR for the nighttime. The reason may be that the tree-based methods are more flexible to identify the non-linear relationships between predictors and $NO_2$" (Lu et al., 2020, p. 7). In order to potentially improve the accuracy for the non-linear models, parameters are tuned. For the random forest model, the number of estimators is set to 1000; the min_samples_split equals 10; the min_samples_leaf equals 5; the maximum features used per tree is set to 4; the maximum depth is 10; bootstrapping is allowed and the random state is set to 42 for reproducibility purposes. For both the lightgbm and xgboost models, the number of estimators are set to 50,000; the reg_alpha equals 2; the reg_lambda equals 0; the max_depth equals 5; the learning rate is 0.0005 and the random state is 42 to ensure reproducibility. Additionally, the gamma of the xgboost model is set to 5. To assess the performance of each of the models, the $R^2$, RMSE and MAE will be examined and compared between the models, as these metrics are a common and useful approach in several studies (Rybarczyk and Zalakeviciute, 2018; Ameer et al., 2019; Chang et al., 2020).

### 2.3.2 Multiple Linear Regression

The variables that were among the most important, identified by the random forest model, are used as predictor variables in the Multiple Linear Regression (MLS) – a statistical technique that uses multiple explanatory variables to predict the outcome of the response variable. However, linear regression techniques can be characterized by complexity and/or over-fitting of the model. To handle such potential pitfalls, the model is extended with the lasso and ridge functions. The ridge regression puts constraints on the model coefficients whereby the penalty term, lambda $\lambda$, regularizes coefficients with large values, consequently reducing the complexity and multi-collinearity. The lasso regression is different in the sense that the penalty equals the sum of the absolute values of the coefficients (Ren et al., 2020). Again the lambda serves as a penalization factor, however, this time stressing the magnitude of the predictors' coefficients, rather than the square of the coefficients. Consequently, coefficients can be equal to 0 which leads to feature selection. For modeling, the alpha is tunned to 0.1, as it leads to the lowest MAE, RMSE, and highest $R^2$ out of options ranging from 0.1 to 1 with a step of 0.1. The equations for the linear regression, error term, ridge regression and lasso regression can be found in supplementary, Equations.

### 2.4 Feature selection

Feature selection is based on examining the Shapley values of each feature (i.e. predictor). The Shapley value for a feature value $j$ is determined by the contribution $\phi_j$ of feature $j$ to the prediction, in this case $NO_2$ concentration levels, compared to the average prediction of the dataset (Shapley, 1953). The contribution of a feature is calculated by examining the difference between the response variable that is obtained when the feature is present with respect to the response variable that is obtained when the feature is absent (i.e. marginal contribution). The contribution of feature $j$, the Shapley value, is calculated via:

$$\phi_j(val) = \sum_{S \subseteq N \setminus \{j\}} \frac{|S|!(|N|-|S|-1)!}{|N|!} (val(S \cup \{j\}) - val(S)) \tag{2}$$





Whereby $N$ describes a set of $n$ features, $val$ is a function that gives the value (i.e. payout) for any set of those features, $S$
corresponds to a subset of features of $N$, $val(S)$ consequently equals the value of the subset (Algaba et al., 2019; Shapley, 1953).

The random forest algorithm is used to determine the influence of each feature. The idea of Shapley feature selection is embodied into two approaches of cross-validation. First of all, the importance of each feature can be based on the mean of
Shapley values with a 10-fold cross-validation. The second approach takes the median rank of the 10-fold cross-validation. Again the ranking of variables of each iteration is based on the Shapley value. The relative positions of each predictor for both approaches can be found in supplementary, figure 8. The Shapley ranking of one fold is visible in supplementary, figure 9.

The median-based approach is chosen to further analyze the preferred number of predictor variables in the model. To de-
termine the preferred number of predictor variables in the model, a random forest algorithm is applied to each number of most influential features, based on the average median ranking, ranging from the two most influential to the thirty most influential features. Thereafter, the RMSE and $R^2$ are used to determine the optimal number of predictors used for modeling.

### 2.4.1 Global

The Number of predictor variables and its related scores are visible in figure 2a, and figure 2b. Based on the different
evaluation methods, having twelve predictors in the model equals to a considerable improvement in terms of R$^2$ and RMSE, compared with using less predictors and a stagnation in model performance when additional predictors are added to the twelve most influential predictors. Therefore, the twelve most important predictors will be included in generating the tree based- and linear models. The most influential predictors on NO$_2$ concentrations, based on the average median are, in sequential order:

population_3000, road_class_3_3000, trafbuf25, population_1000, nightlight_450, nightlight_3150, trafbuf50,
245                 road_class_3_300, bldden100, ndvi, road_class_2_25, and trop_mean_filt_2019.

The twelve predictors are used in the different linear and non-linear models. These models are evaluated on different terms, being R$^2$, MAE, and RMSE. Afterwards, predictions are made for different areas with different demographic characteristics, including two 700,000+ inhabitant cities Amsterdam and Hamburg, Utrecht, which represents a middle-sized city (350,000+ inhabitants), and Bayreuth, which represents a city of a relative low population (70,000+ inhabitants).

### 2.4.2 Local

Due to the poor performances by the random forest both over all the station measurements (supplementary, figure 10a, 10b, and 10c), and per spatial group (supplementary, table 2), the best model is not selected by the random forest algorithm and cross-validated Shapley approach. Rather, the best subset regression is used for variable selection. This approach consists of testing





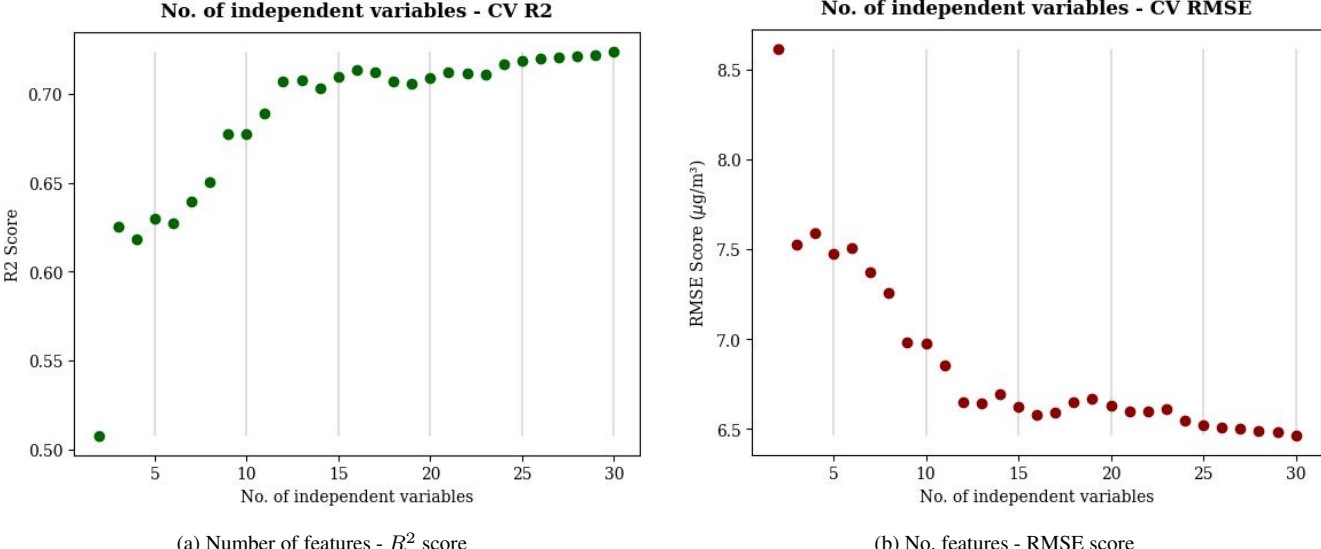

(a) Number of features - $R^2$ score

(b) No. features - RMSE score

**Figure 2.** Ten-fold cross validation: no. features - model performance (global)

all possible combinations of predictor variables, thereafter selecting the model based on some statistical criteria (Kassambara,
2018). The maximum number of predictors considered is equal to 30.

The relevant statistical criteria used are the adjusted $R^2$, CP, and BIC scores. As the preferred number of features may differ
per statistical criterion, additional strategies may be necessary. Kassambara (2018) points to a rigorous approach whereby a
k-fold cross validation is applied to select a model based on the prediction error computed on the new test data. Using the
relationship between number of features and prediction error, eventually nine features are identified that will be used for local
modeling. The nine features, ranked by their importance are:

nightlight_450, nightlight_4950, population_3000, road_class_1_5000, road_class_2_1000, road_class_2_5000,
road_class_3_100, road_class_3_300, trafbuf50

Once the optimal model is determined, a distinction is made between a linear model and a mixed-effects model, whereby a
grouping effect, based on shared spatial characteristics, is integrated. Complementary, ordinary kriging and universal kriging
are applied whereby the automap r package is used for fitting the variograms. As the whole Amsterdam area on a 100 meter
resolution is considered for the prediction area, and nine predictors are included in the universal kriging approach, parallel
computing is applied to address the problem of big computation.





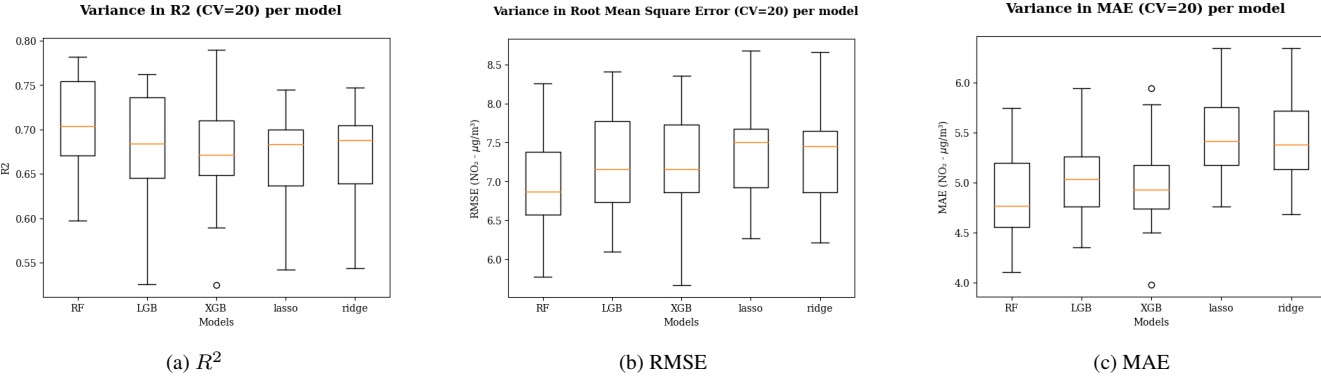

(a) $R^2$  (b) RMSE  (c) MAE

**Figure 3.** The 20-fold cross-validation: performance per model (a) $R^2$, (b) RMSE, (c) MAE (global). The upper- and lower quartiles give an indication of variance; RF = random forest, LGB = lightgbm, XGB = xgboost

# 3 Results

## 3.1 Models

### 3.1.1 Global

Evaluating the different linear- and non-linear models is done by performing a 20-fold cross-validation, thereby examining the variance and mean statistics per model in terms of $R^2$, MAE and RMSE (figure 3a, figure 3b, and figure 3c).

With 20-fold cross-validation, The xgboost model performs better than the lightgbm model in terms of average $R^2$ and MAE, however the RMSE is slightly higher. The linear models (i.e. lasso and ridge) score similarly to the non-linear models, especially in terms of the $R^2$ and RMSE. Generally, the random forest model has the best performances out of the considered global models as the average $R^2$, average RMSE and average MAE is best for this model. The robustness of the random forest model is relatively high too, as the standard deviation is lowest in terms of $R^2$ and RSME (figure 3a and figure 3b).

*Accounting for spatial characteristics*

The comparison is not only made between linear and non-linear models, and global- and local models, but also between spatial groups, to take into account the potential spatial heterogeneity of observations. Meyer and Pebesma (2021, 2022) argue that the increasing popularity of global maps, due to their ability to fit linear and non-linear and complex relationships, are subject to misinformation. A global model that is trained can make accurate predictions as long as global predictors are available, however at places where predictions are far beyond the data, poor predictions by a trained random forest model are not uncommon, given that predictor values of the predicted area do not resemble the training data. Therefore, the potential difference in global model prediction accuracy between different spatial characteristics is examined. An additional variable is



used that describes the spatial characteristic for each observation in the global dataset. Examining the descriptive statistics per spatial group already exposes interesting differences in terms of $NO_2$ concentration levels (table 2).

**Table 2.** Descriptive $NO_2$ statistics for each spatial group, measured in $\mu$g/m3

| Group | Count | Mean | Sd. | Min | 25% | 50% | 75% | Max |
|---|---|---|---|---|---|---|---|---|
| Urban | 85 | 38.865 | 13.065 | 15.768 | 28.172 | 38.076 | 47.923 | 78.882 |
| Low population | 138 | 27.601 | 9.769 | 7.872 | 19.876 | 26.876 | 34.407 | 56.706 |
| Far from road | 259 | 16.653 | 8.341 | 2.122 | 10.331 | 15.892 | 22.518 | 48.887 |

Table 3 describes the performances, in terms of $R^2$, RMSE and MAE, for each spatial group, per model. For each model and for each performance criterion, the observations that are far from roads, perform considerably better than the observations close to roads, for both urban and low population areas. Thereby, the non-linear models outperform the linear models when the data is trained on observations that are far from roads and observations that are close to roads but are characterized in low population areas. For urban areas, the performances between the linear and non-linear methods are less distinguishable

which might be explained by the relatively low number of observations. Due to the relatively limited number of observations and heterogeneous character of data that is apparent in the urban class, ensemble tree-based methods have poor learning conditions, possibly having fewer abilities to learn and discover patterns.





| | | | URB | | | LP | | | FFR | | |
|---|---|---|---|---|---|---|---|---|---|---|---|
| Models | | | $R^2$ | RMSE | MAE | $R^2$ | RMSE | MAE | $R^2$ | RMSE | MAE |
| Non linear | RF | *Mean* | 0.271 | 10.994 | 8.964 | 0.387 | 7.285 | 5.361 | 0.712 | 4.189 | 3.007 |
| | | *SD.* | 0.099 | 1.298 | 0.950 | 0.185 | 1.323 | 0.762 | 0.102 | 0.983 | 0.550 |
| | lightgbm | *Mean* | 0.175 | 11.631 | 9.477 | 0.367 | 7.381 | 5.468 | 0.725 | 4.075 | 2.872 |
| | | *SD.* | 0.145 | 0.226 | 0.955 | 0.226 | 1.513 | 0.739 | 0.120 | 1.040 | 0.537 |
| | xgboost | *Mean* | 0.228 | 11.230 | 9.147 | 0.426 | 7.060 | 5.228 | 0.737 | 3.991 | 2.774 |
| | | *SD.* | 0.150 | 1.014 | 0.807 | 0.183 | 1.340 | 0.687 | 0.116 | 1.096 | 0.530 |
| Linear | ridge | *Mean* | 0.328 | 10.491 | 8.617 | 0.348 | 7.517 | 5.703 | 0.696 | 4.358 | 3.211 |
| | | *SD.* | 0.127 | 1.080 | 0.860 | 0.167 | 1.139 | 0.606 | 0.103 | 1.133 | 0.564 |
| | lasso | *Mean* | 0.265 | 10.936 | 9.017 | 0.282 | 7.859 | 6.047 | 0.613 | 4.912 | 3.749 |
| | | *SD.* | 0.177 | 1.159 | 1.040 | 0.201 | 1.105 | 0.672 | 0.119 | 1.153 | 0.678 |

**Table 3.** Model performance per spatial group (CV = 20). RMSE and MAE are represented in $NO_2$ ($\mu$g/m3) values.

URB: Urban (near road, high population), LP: Low Population (near road, low population), FFR: Far From Road.

*Spatial prediction patterns*


Figure 4 shows the $NO_2$ spatial predictions for the Amsterdam area per model, a-c relating to the non-linear spatial predictions, d and e relating to the linear techniques. Generally, the linear spatial predictions are more discrete, compared to the non-linear techniques. To elaborate, values in the linear predictions are more extreme (i.e. below 15 or above 50) and relatively low and high values tend to be closer to each other, in contrast to non-linear prediction patterns, where the predicted $NO_2$ patterns tend

to be more smooth. Another interesting aspect is the identification of a high $NO_2$ hot spot in the southwestern part of the study area for linear techniques, while this hot spot is not identified by the non-linear predictions. Generally, major roads (highways, national roads) are identifiable and other urban areas (e.g. Haarlem) are characterized by high $NO_2$ concentration levels.

Figures 5 a-f show the spatial patterns of the predicted $NO_2$ concentrations for Hamburg (a and b), Utrecht (c and d) and Bayreuth (e and f) for the random forest and ridge models. Prediction patterns that are related to the ridge models seem to

be more scattered, compared to the random forest prediction maps where patterns are more smooth - a similar finding to Amsterdam where non-linear prediction patterns are smoother than linear techniques. The other models (lightgbm, xgboost, lasso) for Hamburg, Utrecht, and Bayreuth (both zoomed in and out) can be found in supplementary sections figure 11a-c, figure 12a-c, figure 13a-c, figure 14a-e respectively. Comparing between maps of these cities, there are noticeable differences in prediction patterns. A most important finding is that the highest air pollution seems to be situated around major roads

in Hamburg while the urban center accounts for the highest air pollution concentrations in Utrecht. The high correlation between major roads and high air pollution could be reasonable explained considering that Hamburg is the 69th of the most



congested cities in the world (Tomtom, 2021). Interestingly, the highest NO$_2$ concentration levels among highways differ spatially between the random forest and ridge models, for example, the highways in the southeastern and western part of the Hamburg area contain high NO$_2$ levels for the ridge models while a nuanced identification is related to the random forest

prediction for the same area. In the random forest prediction map for Hamburg, air pollution among roads in the center and northern part of the city is more pronounced compared to the ridge model equivalence. Additionally, the magnitude to which high air pollution is related to major roads, is considerably higher for Hamburg, compared to Utrecht and Bayreuth. Still, the relationship between the presence of roads and heavier air pollution concentration is identifiable for both Utrecht and Bayreuth, especially with the ridge model predictions. For Utrecht, the urban center is more pronounced in terms of high NO$_2$

concentration levels, when compared to Hamburg and Bayreuth. Moreover, the ridge model applied on Utrecht identifies more clusters (i.e. scattering) of NO$_2$ values in the periphery while the predicted NO$_2$ values are more scattered in the urban center for the random forest when compared to the ridge model - again, this difference in prediction patterns between a linear and non-linear model is apparent for the Amsterdam area too. Bayreuth is characterized by moderate air pollution and very low (<15 $\mu$g/m3) pollution in rural areas surrounding the city - some clusters exceeding the 15 $\mu$g/m3 benchmark are noticeable

that correspond to other villages in the area, hinting to the influence of population or building density on air pollution (see also supplementary, figure 14a-e).

Figure 6 shows the distribution of predicted NO$_2$ per model for each location. The spatial prediction maps of each city are also shown in bar charts, whereby high air pollution values in the bar charts align with red areas in the spatial prediction maps and vice versa. For the two dutch cities, Amsterdam and Utrecht, the mean of predicted NO$_2$ is higher for the non-linear models. In

contrast, the predicted values are higher for linear models when applied to Bayreuth. The variance in predicted values is higher in linear models for the largest cities (Amsterdam and Hamburg).



(a)        (b)        (c)

(d)        (e)

Predicted NO2 (μg/m³)

| -100 to 0 | 0 to 15 | 15 to 20 | 20 to 25 | 25 to 30 | 30 to 35 | 35 to 40 | 40 to 45 | 45 to 50 | 50 to 100 | 100 to 1,000 |

**Figure 4.** Spatial patterns of predicted NO$_2$ (100m), measured in $\mu$g/m3, per model for Amsterdam - non linear techniques (top): (a) = random forest, (b) = lightgbm, (c) = xgboost; linear techniques (bottom): (d) = lasso, (e) = ridge. Extent = 30km x 30km



(a)                                    (b)                                    (c)

(d)                                    (e)                                    (f)

Predicted NO2 (μg/m³)

| -100 to 0 | 0 to 15 | 15 to 20 | 20 to 25 | 25 to 30 | 30 to 35 | 35 to 40 | 40 to 45 | 45 to 50 | 50 to 100 | 100 to 1,000 |

**Figure 5.** Spatial patterns of predicted NO$_2$ (100m), measured in $\mu$g/m3, per model for Hamburg (extent = 30km x 30km), Utrecht (extent = 25km x 25km) and Bayreuth (extent = 10km x 10km) - top: left = random forest (Hamburg), right = ridge (Hamburg), right = random forest (Utrecht); bottom: left = ridge (Utrecht), middle = random forest (Bayreuth), right = ridge (Bayreuth)





**Figure 6.** Distribution predicted NO$_2$ ($\mu$g/m3) per model and per location

### 3.1.2 Local

The performances of the local models are composed of the R$^2$, RMSE, and MAE, and obtained through a leave-one-out cross-validation approach. With the mixed-effects model, fixed and random effects are included. Fixed effects consist of the most

influential predictors while random effects account for potential spatial trends in the data. The spatial trends in the data related to observations being clustered in a way. The spatial character of the observation, i.e. whether an observation is situated in an urban area, low-populated area, or far from road area, accounts for the random effect in the model. In contrast, the linear model composes all the fixed effects while neglecting the possibility of observation clustering. Additionally, two kriging methods are used for local modeling, being ordinary- and universal kriging. Table 4 shows the model performances for the linear model, the

mixed-effects model, the ordinary kriging model, and the universal kriging model, whereby a leave-one-out cross-validation is applied. The ordinary kriging model shows the poorest performance, which can be explained by it's spatial prediction patterns



(figure 7) and parameters (e.g. a limited range in which other observations are considered). Using auxiliary variables results in a prediction accuracy improvement as the metrics of the universal kriging model are considerably better than the metrics of the ordinal kriging model. However, accounting for spatial autocorrelation does not automatically result in a higher accuracy since the linear model performs better than the universal kriging method. At the same time, accounting for random effects yields a higher $R^2$, a lower RMSE, and a lower MAE.

**Table 4.** Model performance (CV = leave-one-out)

|  | $R^2$ | RMSE ($\mu$g/m3) | MAE ($\mu$g/m3) |
|---|---|---|---|
| ordinary kriging | 0.072 | 8.542 | 7.052 |
| linear model | 0.307 | 7.412 | 5.955 |
| mixed-effects model | 0.326 | 7.315 | 5.808 |
| UK (model + kriged residuals) | 0.277 | 7.749 | 6.097 |

Table 5 shows the model results per spatial group, again based on leave-one-out cross-validation. Same as the results of the global model, the results for the local models indicate that models trained on "urban" observations perform poor - however, the proximity to the road does not necessarily influences the model performance since the $R^2$ of the low population class is higher than the $R^2$ of the far from road class. In contrast to the models trained on the global dataset, which perform best in far from road areas, the models trained on the local dataset perform best in areas that have low populations and have close proximity to roads. An explanation may be that observations in "far from road" areas for the local dataset are more similar to observations in the urban and low-population areas when compared to the global dataset, as the predictor characteristics are more uniform in a relatively small area (i.e. local dataset) in relation to a relatively large area (i.e. global dataset).








**Table 5.** Model performance per spatial group (CV = leave-one-out-cross-validation). RMSE and MAE in $\mu$g/m3

| Models | URB | | | LP | | | FFR | | |
|---|---|---|---|---|---|---|---|---|---|
| | $R^2$ | RMSE | MAE | $R^2$ | RMSE | MAE | $R^2$ | RMSE | MAE |
| ordinary kriging | 0.072 | 8.257 | 6.772 | 0.223 | 8.558 | 6.575 | 0.072 | 9.029 | 8.303 |
| linear model | 0.140 | 7.890 | 6.360 | 0.509 | 6.8 | 5.301 | 0.147 | 7.390 | 6.202 |
| mixed-effects model | 0.141 | 7.874 | 6.316 | 0.524 | 6.505 | 5.298 | 0.115 | 7.404 | 5.644 |
| UK (model + kriged residuals) | 0.161 | 8.068 | 6.27 | 0.487 | 6.938 | 5.174 | 0.037 | 7.19 | 8.299 |

URB = Urban (near road, high population), LP = Low Population (near road, low population), FFR = Far From Road- Mixed-effects model results are identical

*Spatial prediction patterns*

Figure 7 shows the predicted NO$_2$ patterns based on the local dataset. The prediction map of the linear model (a) is fairly
similar to the prediction maps of the mixed-effects (c) model and universal kriging (e) model - the models identify a high NO$_2$
concentration cluster at the northwestern part of Amsterdam. Further examination unravels that this cluster is likely highly
influenced by the predictor road class 2 5000 (i.e. primary roads within 5000m), as this predictor shows a similar cluster at
the same location (supplementary, figure 16, figure 17a-i). Two models which account for the spatial groups first, before the
modeling process, show comparable patterns whereby the influence of roads is clearly visible - be it via the predictors itself,
or the spatial groups (see also supplementary, figure 18). The relative low NO$_2$ values that are along the roads in the outer
Amsterdam area can be explained by the division of the spatial groups. To extend, the presence of predictor values that have a
high standard deviation can influence the NO$_2$ values for that spatial group in such a way, that certain parts of the prediction
area have over- or underestimated prediction values. The patterns that are along the roads belong to the spatial group "low
population" whereby observations within this group are in the vicinity of roads (<100m). Comparing this spatial group, to
the spatial group "far from road", the data distribution for every predictor in low population is substantially different than
the data distribution for every predictor in the group "far from road", leading to different learning patterns which explain the
relative high prediction values along the roads (supplementary, figure 19a-i). At some places, negative predicted values are
apparent albeit few. This is likely a cause of the training dataset having different feature characteristics than the testing dataset.
Comparing the local prediction patterns to the global prediction patterns, a cluster of high air pollution in the northwestern part
of Amsterdam is visible in some local models that is not visible in the global models (as discussed, these models refer to the
general linear-, mixed-effects-, and universal kriging models). A possible explanation for why the cluster is identified in the
local dataset, as opposed to the global dataset, may be the difference in spatial distribution of NO$_2$ values between the local-
and global datasets, resulting in different learning patterns between the local- and global models (figure 1).

Figure 8 summarizes the predicted NO$_2$ value distribution for the linear- and universal kriging methods. Just as the spatial
prediction patterns, the predicted value distributions are fairly similar for the models which account for spatial groups before
modeling on the one hand, and the general linear-, mixed-effects-, and universal kriging models on the other hand. As the
models for every spatial group can predict NO$_2$ considerably different, given the substantial difference in training data per





spatial group, it may not be surprising that many prediction samples classify as outliers, be it correctly or incorrectly. The prediction distributions of the local models without outlier correction can be found in supplementary, figure 20.



**Figure 7.** Spatial patterns of predicted NO$_2$ ($\mu$g/m3) at 100m resolution based on local dataset - top: left = linear model, middle = linear model separating for spatial groups, right = mixed-effects model; bottom: left = ordinary kriging, middle = universal kriging, right = universal kriging separating for spatial groups







**Figure 8.** Distribution predicted NO$_2$ ($\mu$g/m3) per model (local) with outliers correction. LR = linear regression, LRsp = linear regression accounting for spatial groups, MEM = mixed- effects model, UK = universal kriging, UKsp = universal kriging accounting for spatial groups, OK = ordinary kriging



*Model comparison*

Figure 9 shows the correlation in predicted $NO_2$ values for the local- and global models, as well as the mobile $NO_2$ map of Kerckhoffs et al. (2019) which is used as benchmark. Some outliers are present in the local linear model that accounts for spatial groups and the universal kriging model that accounts for spatial groups, resulting in a relatively low correlation with
the open $NO_2$ dataset. Therefore, outliers are removed to enhance the correlation between mentioned models on the one hand and the $NO_2$ dataset on the other. There are a plethora of outlier detection methods, however multiple techniques such as chi-squared test, Dixon test, Grubbs test, and 1.5 times IQR, label values that would be within the IQR of other models, as outliers. Therefore, a manual threshold is chosen which is based on the data distributions of other models. Since the maximum value of the ten models (excluding the two where the outlier detection is applied on in the first place) is 85, the outlier detection
threshold will be tuned to 85 as the upper bound. For the lower bound, negative values are considered as outliers. Afterwards, the outlier filtering is applied to all models, potentially filtering out negative prediction values for other models. The new correlation matrix is visible in supplementary, figure 21. Furthermore, global models have a high correlation with each other; interestingly the lasso model has lesser prediction pattern similarities with the other global models. The ordinary kriging model has low similarity with other models which may not be surprising, given the model settings (e.g. a low searching radius) and
prediction pattern. Comparing the models to the mobile $NO_2$ map, the local models generally show more similarity than global models. A reason for this may be that both the local dataset and the mobile $NO_2$ map are projected on the Amsterdam area.



**Figure 9.** Comparing model predictions whereby the numbers equal the Pearson correlation coefficient. RF = random forest, LGB = lightgbm, XGB = xgboost, LR = linear regression, LRsp = linear regression accounting for spatial groups, MEM = mixed- effects model, UK = universal kriging, UKsp = universal kriging accounting for spatial groups, OK = ordinary kriging, no2 = mobile NO₂ map.



## 4   Discussion

Accurate modeling and estimation of $NO_2$ concentration levels is essential to get a better understanding of air pollution, thereby deducing political implications that can encourage a healthy society. Several important findings of this research, related to this,

are discussed below.

*Relationship between predictors and other pollutants*

For both the global and local datasets, variables concerning traffic and population density are selected among the most in-

fluential predictors, a finding that is similar to the statements of Beelen et al. (2013) as they argue that including such variables encourages prediction accuracy. Additionally, the high influence of traffic on $NO_2$ concentrations match the findings of Lu et al. (2020) and Chen et al. (2019). Chen et al. (2019) moreover find that the most important feature(s) differ per pollutant measured. As this research only predicts $NO_2$ concentrations, it will be interesting to see to what degree choosing a different pollutant results in different models and prediction patterns. For instance, major roads and highways are clearly identifiable in

Amsterdam, Bayreuth, Hamburg, and Utrecht, hinting at the high influence of traffic on $NO_2$ concentrations. As other features may be chosen as the highest influencers on pollutants other than $NO_2$, prediction maps are subject to change too, potentially leading to substantially different prediction maps for CO, $O_3$, PM2.5, PM10, and $SO_2$. Examining to what degree modeling and prediction patterns differ for different pollutants is therefore important to better understand the relationship between space and health, especially since the impact of air pollution on health may differ per pollutant (He et al., 2022).


*Addition of temporal analysis*

The temporal analysis in this research is limited and can be extended. For instance, Lu et al. (2020) account for day and night pollution. Complementary, a distinction between week- and weekend days can be executed, and the influence of certain

events, such as COVID, could influence the concentration levels of pollutants. Accounting for those temporal aspects potentially influences the feature importance, modeling, prediction maps, and prediction quality per spatial group. For instance, Wong et al. (2021) state that the atmospheric environment is substantially improving during the COVID-19 pandemic - a period in which urban mobility is virtually zero. The importance of time on air pollution predictors such as traffic and air pollution itself is demonstrated by Kendrick et al. (2015) too, as they argue that $NO_2$ has seasonal and diurnal variation as a function of

traffic volumes alongside a major arterial, and Minoura and Ito (2010) by stating that traffic signals are related to roadside air quality in Tokyo, Japan. Therefore, future research should implement temporal models, for instance, ARIMA and its relatives (Wong et al., 2021), or apply spatiotemporal models (van Zoest et al. (2020)) to examine the effect of time or space-time on feature influence and model accuracy, not only between global and local datasets, but also between spatial groups (e.g. urban, low population, far from the road) and administrative regions such as cities.




*Accounting for spatial groups*

Considering the global dataset, the differences between the linear and non-linear techniques are barely noticeable. Although the random forest model generally performs best (highest $R^2$, lowest MAE), the $R^2$ of the ridge is higher than the lightgbm and
xgboost models. However, when accounting for the spatial groups - urban, low population, and far from roads - the differences in model performance between linear and non-linear techniques become more distinguishable, whereby the latter techniques perform better. This is especially true for observations far from roads, where data generally is more homogeneous, while the model performances between linear and non-linear techniques in urban areas are less pronounced, albeit both techniques perform poorly here. So, in the first instance, this study does not acknowledge better prediction performances by the non-linear
techniques, as suggested by Weichenthal et al. (2016), Reid et al. (2015), Chen et al. (2019), and Lu et al. (2020), however when accounting for spatial characteristics, non-linear techniques perform considerably better than linear techniques, especially in more homogeneous areas which support the arguments made by Meyer and Pebesma (2021) that non-linear predictions are of higher quality in areas that have similar environmental variables as the training data. Complementary, the more heterogeneous data nature of urban areas results in converging performances between linear and non-linear techniques. Thereby, both tech-
niques perform poorly which, when not controlling for spatial characteristics of the observations, may have been unnoticed. The poor prediction accuracy in urban areas is worrisome given that the impact of air pollution can depend on the surrounding environment, i.e. people that live in the vicinity of traffic-heavy roads (which are often more present in, or around urban areas) and/or industries facing higher exposure to air pollution (He et al., 2022). Though spatial grouping greatly improves the predicting reliability, it can present patterns that are counter-intuitive. For instance, in some areas, the predicted $NO_2$ concentration
levels are lower along roads than the concentration levels of the rural surroundings.

*Spatially varying on feature importance*

While the feature importance is equal between cities, the influence of predictors on $NO_2$ concentrations differs between the
case cities. For instance, building density and population are more prevalent contributors to air pollution in Utrecht, compared to Hamburg, while traffic has a higher influence on high $NO_2$ concentrations in Hamburg, compared to Utrecht. Therefore, it is suggested to further examine the effect of space on predictor influence to stress the importance of not only global and national but also local policies in managing air pollution. Additionally, local and global models are applied to different cities but with the same predictors. As the case cities unravel that high $NO_2$ can be attributed to different predictors per city, applying models
with different features may yield better prediction results. An important condition is of course that there should be enough observations for every city to make this scenario possible to avoid unreliable predictions.

*Model quality*

The limited number of observations in the local dataset poses a problem in making accurate predictions, especially for non-





linear techniques. Consequently, six different models are chosen for the local dataset. Interestingly, accounting for spatial autocorrelation does not equal model improvement while accounting for spatial groups does. Currently, outliers are omitted after the model predictions to deal with unreliable predictions. Transforming the original data could avoid data out of range (e.g. $< 0$ mg/m$^3$). In our study, such transformation, e.g., a log transformation, is not applied, however, the application could

have yielded a better data distribution of predictors, potentially leading to prediction improvements. Moreover, while the lasso and ridge models seem to be useful with the global dataset, the predictions are unsatisfactory with the local dataset. Given that the "road class 2 5000" has relatively high values and is of high influence on the NO$_2$ concentration levels in the local modeling, using a lasso and/or ridge model would have been useful. Also, important model features may be analyzed in more detail to get a deeper understanding of air pollution. In the models of this study, traffic is a prevalent feature, however, no

distinction is made between traffic type (e.g. cars, buses, trucks), car type (e.g. electric, diesel), and distance traveled while such aspects can be influential to air pollution, as suggested by Wong et al. (2021). Performing a deeper analysis of features may unravel interesting findings that would otherwise have been unnoticed, for instance, distinguishing between vehicle types may show that relatively many trucks are on specific roads (e.g. going to or from the port of Hamburg) which might explain certain clusters of high NO$_2$ concentration levels at certain places.


*The relationship between air pollution and socioeconomic status*

As the relationship with socioeconomic status is not evaluated in this study, future research should examine the relationship between linear and non-linear predictions with socioeconomic status so that not only the quantification of exposure but also the

assessment of inequality in health. For instance, Jiao et al. (2018) argue that the relationship between socioeconomic factors and the health environment is understudied. Additionally, Bai et al. (2019) mention that very few studies have analyzed the influences of air pollution from the perspective of meteorological factors, pollution sources, and socioeconomic status. Next to the missing link with socioeconomic status, the temporal aspect is important when examining air pollution and society, and human exposures to air pollution (Lu et al., 2020).

**5 Conclusions**

In this study, we compared various statistical models for predicting NO$_2$ concentrations at different scales (local vs. global) for their spatial and overall prediction accuracy, as well as NO$_2$ maps of cities with distinctive characteristics such as population, and the importance of features. One of the key findings of this study is that the model performance varies little with models of different levels of complexity, but spatially due to spatial heterogeneities in traffic and urban features. Non-linear techniques

predict better in areas far from roads and in areas close to roads but with low population density, compared to linear models. Additionally, global model prediction accuracy is considerably higher in areas far from roads than in areas close to roads. Furthermore, population density is associated with global model prediction accuracy, whereby low-populated areas yield higher prediction accuracy than high-populated areas. In contrast, methods preferred in global modeling appear to be unfavorable in



local modeling. Local models show that low-populated areas that are close to roads yield higher prediction accuracy than high-populated areas close to roads or areas far from roads. The relatively few NO$_2$ observations used in the local models could explain why non-linear models perform poorly. However, lasso- and ridge models also do not increase the prediction accuracy. The prediction accuracy in spatial groups differs from the global models. We also found that modeling the spatial autocorrelation does not contribute to the improvement of the local modeling accuracy, but accounting for spatial groups does. Lastly, non-linear methods predict smoother patterns compared to linear methods in some cities, and different modeling techniques lead to different NO$_2$ clusters in the prediction map. These results suggest that only looking at the prediction accuracy is not sufficient for evaluating the statistical models in air pollution prediction.

*Code and data availability*

Available via: https://github.com/FoekeBoersma/A-close-look-at-using-national-ground-stations-for-the-statistical-mapping-of-NO2 and https://doi.org/10.5281/zenodo.8397133

Datasets larger than 100MB are included and can be accessed in another repository via: https://doi.org/10.5281/zenodo.7948161

*Author contributions.*

Conceptualization, F.B. and M.L.; methodology, F.B. and M.L.; validation, F.B.; formal analysis, F.B.; investigation, F.B.; resources, F.B. and M.L.; data curation, F.B.; writing—original draft preparation, F.B. and M.L.; writing—review and editing, F.B. and M.L.; visualization, F.B.; supervision, M.L.; project administration, F.B. and M.L.; funding acquisition, F.B. and M.L. All authors have read and agreed to the published version of the manuscript.

*Competing interests.*

The authors declare that they have no conflict of interest.



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
