# Peer review of "A close look at using national ground stations for the statistical modeling of $NO_2$"

_EGUsphere, 2023_

## Referee Comment (RC2)

**Introduction**

In this paper, the authors aim at discussing the role of spatial heterogeneity in spatio-temporal prediction of ground-level air quality (i.e., airborne pollutant concentrations) using both a global and a local approach. The authors refer to a global dataset consisting of all the ground station measurements in Germany and the Netherlands, and to a local dataset comprising only the ground monitoring station in the Amsterdam area. The authors attempt to assess the performance of several algorithms across different spatial scales (global and local) and validate the predictive accuracy when ignoring and when considering local spatial characteristics (i.e., density and population density). The main findings state that that the model performance strongly depends on the considered spatial scale and on the considered spatial locations.

The paper addresses the issue of spatial prediction of air quality in a very broad way and tests several interesting dimensions. However, the work done and the methodology are not rigorous and have several critical points. In particular, several methodological inaccuracies, poor analytical rigor and unclear (if not unwarranted) choices emerged during the reading. Therefore, I suggest that the paper should not be accepted in its present form and should be subject to major revisions (especially in methodology).

**General comments:**

Hereafter, I state my major concerns that need to be addressed and clarified.

- The overall readability of the text is very poor:
  - the presentation of machine/statistical learning models is very superficial (there are no formulas and no technical modeling aspects are discussed);
  - names and acronyms are inserted into the text without appropriate discussion and description;
  - many sentences need to be rewritten as they are unclear;
  - the paper is very long and confusing: reading requires continuous jumping from one section to another to understand what models and assumptions the authors are analyzing.
- Methodology:
  - Models are presented without specifying their technical characteristics, differences and rationale for their use. No formulas explaining the structure of the models (e.g., the spatiotemporal structure of random effects) are included by the authors. A paper using statistical methodology should never assume that the reader is aware of the methods;
  - Machine learning models (e.g., random forests, xgboost and lightgbm) require great caution and understanding before their use. They are (partially) black-box models with attributes devoted to prediction rather than interpretation of phenomena (while they greatly improve predictions, they also make the results lose interpretive meaning and risk becoming tools that cannot be used by policy makers or practitioners);
  - The expression "linear models" denotes the class of models that are linear in their parameters. It is a very large family. Personally, I struggled to understand which linear models you considered: linear regression? at what scale (original or logarithmic)?

ridge and LASSO are linear, but differ significantly from pure OLS because of the penalty;

- o Transformations and distribution of data: the text does not mention the problem of positive skewness (typical of the airborne pollutant concentrations world) (Mudelsee). Clearly, the prediction changes considerably if transformations (e.g., logarithm) are applied to adjust for skewness or if general models with non-Gaussian distributions (e.g., GLMs and GAMs) are used. Where do you stand with respect to this problem?

- o There are dozens of linear models and spatio-temporal mixed-effects models in the literature that provide a fair trade-off between interpretability and predictive ability. In the text, none of them are mentioned. I do not intend to cite specific ones, but just type in Google scholar "spatio-temporal models" to retrieve them. I would suggest starting with the spatio-temporal modeling of Wikle-Cressie (who have made history in this branch of research) and colleagues [1, 2];

- o The use of cross-validation is relevant for assessing the predictive ability of models. However, remember that random K-fold, as well as LOOCV, are studied for the cross-sectional world. In a spatiotemporal context, they require ad-hoc adjustments that preserve the correlation structure in time and space. In this regard, see the work of Meyer-Pebesma [3-8] on the role of spatial CV and how it is the ideal substitute for random K-fold in the context you study. Also, for the sake of completeness, I suggest adding the word "out-of-sample performances" every time you use CV because it must be clear to reader that all the metrics are computed in a training-test framework to assess predictive capacities of model (and not in-sample fitting);

- o Feature selection involves an overwhelming number of alternative techniques: the authors use Shapley values, variables importance, penalty (lasso and ridge), bust subset, etc. Ideally, only one method should be chosen to select relevant covariates so that model results are comparable and not data-dependent. Similarly, it is somewhat problematic to have two or more CV schemes (20-fold and LOO) that prevent proper comparison of the models;

- o When reading, I had the feeling (probably wrong) that there was a misunderstanding of the statistical tools used. For example, box plots do not assess the "variance" but the "variability" of a phenomenon and use the median (not the mean) as a reference point, as it is robust to the presence of outliers (frequent in air quality).

**Specific comments and technical corrections:**

- Abstract, line 2: "model and predict air pollution over space and time"
- Section 2.1, page 5, row 143: the authors state that "We used the precipitation from weather stations (National Centers for Environmental Information, 2017)". Why not directly using Copernicus ECMWF ERA-5 data, which naturally cover the whole Europe with a fine scale (compared to the study area)? Since you used spatial interpolation (ordinary kriging), how do you account for the interpolation uncertainty generated by this approach? How does it reflect on the following stages?
- Section 2.1, formula after row 165: please, explicitly define the symbols/quantities used in the formula. In its current form, it's not easy to understand how the traffic is computed;
- Section 2.2

- o Row 171: The average median of what? which values are you considering to rank the features? (later on we discover that you take the median of the rankings… but here it is not clear)
- o Rows 175-177: Being the first time you cite LASSO, lightgbm and xgboost models, I suggest using the extended names followed by the acronyms. Also, add some theoretical references on the models (e.g., papers explaining the full methodology);
- o Row 178: for an extensive comparison in assessing the spatio-temporal prediction accuracy of tree-based methods, linear mixed models and geostatistical mixed models I suggest the papers from the Fassò research group [9-12];
- Section 2.3
  - o Actually, the considered models are only described in words but it is difficult to compare it from the analytical perspective. I suggest adding a synoptic table which synthesize the characteristics of the considered models. For instance, the table could state if a model explicitly considers (or not) spatial, temporal or spatio-temporal components (e.g., spatial random effects), if a model is penalized or not (e.g., LASSO), if a model includes covariates or not (e.g., ordinary kriging);
  - o Section 2.3.1: can you consider including the recent works on spatial random forests, which extend classical RF to a spatial prediction context [13]? As the aim of the paper is to assess the spatial prediction accuracy of models, this new class could improve a lot your findings;
  - o Section 2.3.2, row 214 ($\alpha$): Please, explicitly define the parameter alpha. Also, if alpha refers to the elastic net mixing parameter, with $0 \le \alpha \le 1$, then you are considering the elastic net penalization, which is a combination of LASSO and ridge, and not exactly LASSO or ridge;
- Section 2.4
  - o Row 224: "N describes a set of n features" is not clear. which is the difference between N and n? What do you mean by payout? Is it the prediction with the N features? Is S the cardinality of the subset of N?
  - o Rows 228-229: please, consider rephrasing the whole sentence: the current sentence seems to state that in general/typically Shapley values are embedded in the two alternative CV approaches. However, this seems to be one of your proposals;
  - o Rows 239-242: Please, consider rephrasing the whole sentence as currently it is confused. My interpretation of Figure 2 is that a sensible/remarkable prediction accuracy improvement is obtained when considering at least 12 predictors. However, the improvement is marginal considering more than 12 covariates (the curves become flat);
  - o Section 2.4.2 (best subset regression): usually, best subset is used in a linear regression framework. Still, it is not clear to me if you are considering its application in linear or non-linear (in this case, which model?) models. Also, best subset regression is typically affected by computational inefficiency as it requires the computation of 2^k-1 models (where k=30 in the number of covariates). Do you have any insights about the computational burden of this step?
  - o Section 2.4.2 (linear models): still, it is not clear to me which class of mixed-effects models are you considering. Please, can you state (in Appendix or Supplementary Materials) the exact formulae and parameter specifications (i.e., which is the structure of the random effects? Are they i.i.d. sequence of Gaussian RVs or are spatio-temporally structured?). Also, later on you state that the "... linear models (i.e., LASSO and ridge) ...": why not considering a multiple linear regression without

penalization? This last model should be directly comparable with penalized approaches.

- Figure 3 (caption): I would say that the upper and lower whiskers provide information about the overall variability of the estimates rather than "variance". Box-whisker plots are typically computed using the IRQ-rule, that is, the whiskers are +-1.5 x IQR (interquartile range, i.e., X_0.75 - X_0.25). Also, box-plots typically use the median as central value. Is the orange line the median? If so, why do you talk about "mean statistic" in the first paragraph of Section 3.1.1? In air quality statistics there is a huge difference among robust (median) and non-robust (mean) methods for assessing the centrality of air quality distributions.

- Section 3.1.1
  - row 286: what do you mean by "uncommon"?
  - right before Table 2: Spatial characteristics is a fundamental feature in air quality statistical modeling. Indeed, local air quality is substantially affected by local weather and environmental conditions. However, why did you not include such variable in the features selection stage? You should be sure about the effective predictive capacity of such variables before including it "a priori". Also, I suppose you used the information through a set of dummy variables (I guess 2 vars). Is that correct? Which one did you choose as reference category? Otherwise, did you used separated/independent models by category (i.e., you estimated all the previous models only for Urban and then for low pop and then for far from road)? In the latter case, you should compare the results with the full dataset very carefully as in the sub-models you are ignoring a large part of the information contained in the full data;
  - row 302: please, clearly state the definition of "more discrete" outcomes or models;
  - rows 307 on: whenever you cite a specific place (e.g., Harleem), please make sure that the area is recognizable on the maps. Where is Harleem in Figures 4 and 5? The same comment holds for all the other cities/locations.

- Figures 4 and 5: they compare different models for different locations. Can you justify this choice? It seems an unfair comparison: to understand the effects of models one should compare different models at the same locations. The comparison you propose is meaningful only if you are sure that, independently on the local conditions, the predictions are comparable (thus there is no spatial effect) and the only relevant factor is the model's definition;

- Figure 6: still, if you use box-plots, then the central value you are comparing is the median. Also, it is not clear to me what you are representing on the box-plots. Are they the distribution of the estimated NO2 concentrations at every point (if so, how many points did you interpolate?) in a specific area (Figures 4 and 5) or are they temporal predictions at some locations (in this case, which locations?) or are the spatio-temporal predictions? Also, where are the results associated with linear mixed models?

- Section 3.1.2: why did you move from a 20-fold CV for global model assessing to a N-fold (LOO) CV for local models? This choice introduces some issues when comparing models as the predictions are computed using sample sizes;

- Table 4: Where are the machine/statistical learning results (i.e., lightgbm, xgboost, random forest)? What is "linear model" and which is its relationship with ridge and LASSO? Why do you compare a different set of models? As for the mapping, if different models are used to compare local and global modeling, the comparison will be biased and unfair;

- Figure 9: are Kerckhoffs's data used to train the models? Why not using the actual NO2 observations (used as response variable of the models) as benchmark? Also, I would plot the

original NO2 concentrations used as response variables in the models. They are the actual benchmark.

**Essential bibliography**

1.    Wikle, C.K., L.M. Berliner, and N. Cressie, *Hierarchical Bayesian space-time models.* Environmental and Ecological Statistics, 1998. **5**(2): p. 117-154.
2.    Wikle, C.K., A. Zammit-Mangion, and N. Cressie, *Spatio-temporal Statistics with R*. 2019: Chapman and Hall/CRC.
3.    Meyer, H., C. Milà, and M. Ludwig, *CAST: 'caret' Applications for Spatial-Temporal Models*. 2022.
4.    Meyer, H., et al., *Importance of spatial predictor variable selection in machine learning applications – Moving from data reproduction to spatial prediction.* Ecological Modelling, 2019. **411**: p. 108815.
5.    Meyer, H., et al., *Improving performance of spatio-temporal machine learning models using forward feature selection and target-oriented validation.* Environmental Modelling & Software, 2018. **101**: p. 1-9.
6.    Meyer, H. and E. Pebesma, *Machine learning-based global maps of ecological variables and the challenge of assessing them.* Nature Communications, 2022. **13**(1): p. 2208.
7.    Milà, C., et al., *Nearest neighbour distance matching Leave-One-Out Cross-Validation for map validation.* Methods in Ecology and Evolution, 2022. **n/a**(n/a).
8.    Meyer, H. and E. Pebesma, *Predicting into unknown space? Estimating the area of applicability of spatial prediction models.* Methods in Ecology and Evolution, 2021. **12**(9): p. 1620-1633.
9.    Fassò, A., et al., *Agrimonia: a dataset on livestock, meteorology and air quality in the Lombardy region, Italy.* Scientific Data, 2023. **10**(1): p. 143.
10.   Maranzano, P., P. Otto, and A. Fassò, *Adaptive LASSO estimation for functional hidden dynamic geostatistical model.* Stochastic Environmental Research and Risk Assessment, 2023.
11.   Maranzano, P. and M. Pelagatti, *Spatiotemporal Event Studies for Environmental Data Under Cross-Sectional Dependence: An Application to Air Quality Assessment in Lombardy.* Journal of Agricultural, Biological and Environmental Statistics, 2023.
12.   Otto, P., et al., *Spatiotemporal modelling of PM $\_{2.5}$ concentrations in Lombardy (Italy)--A comparative study.* arXiv preprint arXiv:2309.07285, 2023.
13.   Patelli, L., et al., *A path in regression Random Forest looking for spatial dependence: a taxonomy and a systematic review.* arXiv preprint arXiv:2303.04693, 2023.

---

## Author Comment (AC1)

**Major revision**

Monday 25 march 2024

Foeke Boersma & Meng Lu

**We appreciate your time and effort into the feedback given. There are a lot of good suggestions which helped us improve the quality of the content. Our replies to your feedback is expressed in bold font.**

**Feedback – reviewer 1**

General comments:

Currently, many studies are using spatially sparse fixed-site measurements to map air pollution on a large scale, ignoring the local spatial heterogeneities such as the intra-city variations. This article evaluated the performance of various algorithms across different scales and validated the accuracy separately in subsets categorized by road density and population density. They found that the model performance varied significantly at different spatial locations. The pattern was found to be different in "global" and local models. The comparison between "global" and local models in terms of intra-city distribution patterns is valuable. However, in its present form, I cannot recommend the article for publication. With substantial revision and restructuring, this article could be a useful addition to the existing literature.

The writing needs further improvement. The current version is not easy to read. First, this is too long. I appreciate the solid work of the authors. But please simplify the main text and consider moving some descriptions/figures to the Appendix. Keep only the core story in the main text and make sure the primary findings and the most important messages stand out. Second, consider restructuring the method/data and the result section. Third, the caption of figures and tables needs more details, including the unit of $NO_2$. Fourth, Clarify definitions like "Far from road" vs "Rural". Last, please pay attention to the tense usage.

Specific comments:

1. The mobile measurements from Kerckhoffs et. al., 2019 were measured on the road. How can they validate the accuracy for the "far from road" group? Did you perform any adjustments?

**Reference to Kerckhoffs is relevant as the study area entails the area of Amsterdam as well (i.e. similar area of interest)**

2. Table 1 describes the predictor features. Why not include land use proportions? Land Use Regression models are efficient and well-accepted methods in air pollution modeling.

**Thank you for your comprehensive and useful feedback and suggestions. Considering your suggestion about including land use proportions in our predictor features, we believe that we already incorporated the land use proportions via various buffer analyses for variables including industry, building density, and road types. In case we**

**misinterpreted this suggestion, please let us know so that we potentially can make the necessary changes.**

3. Figure 4. It would be better to plot the map of differences between the model tested and the benchmark (i.e., NO2 estimations from Kerckhoffs et. al., 2019). I would be curious about the difference in spatial distributions between the "global" and local models.

4. A restructuring of the data/method section is recommended. Begin with the introduction of the data source, ensuring clarity on the source of the population information and road class when discussing spatial groups. Consider adding a table summarizing model input/algorithms for ease of understanding. Move some algorithm introductions to the appendix.

**I changed the structure of the methodology. In the renewed methodology, the data (2.1) is introduced first, followed by a short elaboration per model used in the next subchapter (2.2); subsection 2.3 elaborates on feature selection which is an important part as it determines the relevant variables for the modeling; the last section of the methodology provides insights into how the models are evaluated and used, thereby showing the relevant models in a table overview.**

5. Please explain why 20-fold cross-validation?

**We select a 20-fold cross-validation to encourage stable estimate; added this sentence in main text.**

Technical corrections:

I have listed some specific points. But not limited to them.

Abstract:

The abstract attempts to encompass numerous findings but allocates insufficient space to elucidate the methodology and experimental setting. A substantial rephrasing of the abstract is needed.

Line 1-5, toing and froing, can be simplified.

Line 6, please provide more details about the meaning of "spatial heterogeneity" in this context.

**Added: "in which characteristics and/or phenomena may change over space"**

Line 9-10 what is the local and global model? Define first, before using it.

**Elaboration given:**

**"Local models are based on the Amsterdam area and perform predictions on the same area. Global models are based on observations throughout Germany and The Netherlands while predictions apply to several smaller
areas of interest in Germany and The Netherlands"**

Methodology:

Line 100-105, not clear. How do you divide the area? Purpose? What is the time frame of these national measurements? Frequency of measuring? Any preprocessing? More details are needed

here. How do you define the less densely populated area? What is the source of the population density data?

Line 121, "rural"= "Far from roads"? Please keep the terminology consistent. Changed to "far from roads"

**Changed to "far from roads"**

Line 123, the label of models should be provided as the legend in the figure instead of in the caption.

Line 130-135, unit of $NO_2$ is missing. This paragraph is not informative. The values can be integrated into the figure 1.

**Removed alinea**

Line 145, More details about kriging and accuracy are needed.

**Added a supplementary, equations and supplementary, parameters containing more details. Reference in main text is added too.**

Line 160-165, is the traffic volume used as the annual average? "The traffic volume is expressed in average hourly traffic" > "The traffic volume is expressed in average hourly traffic, measured over the year 2017." Table 1. it would help readers to understand the data distribution by adding columns such as numbers and some statistics like mean, median, quantiles etc.

**Removed table 1, however added a table with descriptives of predictors that are used to classify spatial groups (Table 1. Descriptive statistics for each relevant variable in the determination of spatial groups for the local- and global datasets)**

Line 168, the section title should begin with a capital letter, and further refinement is necessary in terms of formatting.

Line 190, not clear. Please do not refer to the citation but to the dataset you have described in section 2.1. Removed this line, also with the intention to limit the main text.

**Removed this line, also with the intention to limit the main text.**

Line 195, rephrase please instead of a direct quote. Removed quote, also with the intention to limit the main text.

**Removed quote, also with the intention to limit the main text.**

Line 196, details of the tuning strategy are missing.

**Added a supplementary, equations and supplementary, parameters containing more details. Reference in main text is added too.**

Result and discussion:

Line 465, how do you compare the influence of predictors between cities? The feature importance is a relative value. The magnitude is not meaningful when compared to the other models.

**Actually by looking at the prediction patters. For Hamburg, high clusters of NO2 seem to coincide more with traffic related variables and less with population related variables; for Utrecht, high clusters of NO2 seem to coincide more in the city center itself (higher building density; population) than traffic related variables (high NO2 clusters along highways are less visible here).**

Line 515, which is opposite to the common knowledge (see Hoek et. al., 2008). Can you explain why non-linear model predictions were smoother?

**Based on this, changed:**

**"The spatial prediction patterns show that non-linear methods generally predict more smoothly than linear methods. Additionally, clusters of predicted air pollution differ within and between cities." → Lastly, non-linear prediction patterns seem to be less prone to overfitting compared to linear methods, and different modeling techniques lead to different NO${_2}$ clusters in the prediction map. (conclusion)**

**"The spatial prediction patterns show that non-linear methods generally predict more smoothly than linear methods." → The spatial prediction patterns show that non-linear methods generally are less prone to overfitting than linear methods. (abstract)**

**"Generally, the linear spatial predictions are more discrete, compared to the non-linear techniques. To elaborate, values in the linear predictions are more extreme (i.e. below 15 or above 50) and relatively low and high values tend to be closer to each other, in contrast to non-linear prediction patterns, where the predicted NO2 patterns tend 305 to be more smooth" → Generally, the linear spatial predictions seem to be more prone to overfitting as these prediction maps are characterized by a higher share of extreme values compared to non-linear techniques (i.e. below 15$\mu$g/m3 or above 50$\mu$g/m3). (section 3.1.1)**

**Feedback – reviewer 2**

Introduction In this paper, the authors aim at discussing the role of spatial heterogeneity in spatio-temporal prediction of ground-level air quality (i.e., airborne pollutant concentrations) using both a global and a local approach. The authors refer to a global dataset consisting of all the ground station measurements in Germany and the Netherlands, and to a local dataset comprising only the ground monitoring station in the Amsterdam area. The authors attempt to assess the performance of several algorithms across different spatial scales (global and local) and validate the predictive accuracy when ignoring and when considering local spatial characteristics (i.e., density and population density). The main findings state that that the

model performance strongly depends on the considered spatial scale and on the considered spatial locations.

 The paper addresses the issue of spatial prediction of air quality in a very broad way and tests several interesting dimensions. However, the work done and the methodology are not rigorous and have several critical points. In particular, several methodological inaccuracies, poor analytical rigor and unclear (if not unwarranted) choices emerged during the reading. Therefore, I suggest that the paper should not be accepted in its present form and should be subject to major revisions (especially in methodology).

**General comments:**

Hereafter, I state my major concerns that need to be addressed and clarified.

• The overall readability of the text is very poor:
o the presentation of machine/statistical learning models is very superficial (there are no formulas and no technical modeling aspects are discussed);

**To make the paper more readable, the formulas are added in the supplementary material. A reference to the equations and technical aspects of the considered models is now available in supplementary, equations.**

o names and acronyms are inserted into the text without appropriate discussion and description;

**lightgbm, xgboost, and lasso are now first written out before referring to the acronyms/names.**

o many sentences need to be rewritten as they are unclear;

check
o the paper is very long and confusing: reading requires continuous jumping from one section to another to understand what models and assumptions the authors are analyzing.

**I changed the structure of the methodology. In the renewed methodology, the data (2.1) is introduced first, followed by a short elaboration per model used in the next subchapter (2.2); subsection 2.3 elaborates on feature selection which is an important part as it determines the relevant variables for the modeling; the last section of the methodology provides insights into how the models are evaluated and used, thereby showing the relevant models in a table overview.**

• Methodology:

o Models are presented without specifying their technical characteristics, differences and rationale for their use. No formulas explaining the structure of the models (e.g., the spatiotemporal structure of random effects) are included by the authors. A paper using statistical methodology should never assume that the reader is aware of the methods;

**Information on technical characteristics, differences and rationales for modeling can now be found in supplementary equations and supplementary parameters. The temporal aspect is neglected in the models unfortunately, as this is outside the scope of this**

**research, however should be addressed in future research.**

o Machine learning models (e.g., random forests, xgboost and lightgbm) require great caution and understanding before their use. They are (partially) black-box models with attributes devoted to prediction rather than interpretation of phenomena (while they greatly improve predictions, they also make the results lose interpretive meaning and risk becoming tools that cannot be used by policy makers or practitioners);

**Thank you for the comment. We agree that tree-based machine learning models require great caution and understanding before their use. However, despite that classical statistical methods such as standard linear regression (linear regression without penalty) have clear interpretations on the parameters, a correct interpretation also depend on the model assumptions. That is, if the true model is highly non-linear and a linear model is used, the standard linear regression can also lose interpretive meaning and risk becoming tools that are not usable. Models such as ensemble trees, besides their potential predictive power, can be interpreted using for example marginals and permutations, also, the uncertainties can be assessed. We admit that many desirable properties they don't have, and that is a main reason that in our study, we compare them with statistical models such as Lasso, standard linear regression, and Kriging.**

o The expression "linear models" denotes the class of models that are linear in their parameters. It is a very large family. Personally, I struggled to understand which linear models you considered: linear regression? at what scale (original or logarithmic)? ridge and LASSO are linear, but differ significantly from pure OLS because of the penalty;

**Thank you for your question. By linear model we mean the model with a linear relationship between predictors and response,**

**Y= Xß**

**We see Lasso and ridge as a general form of linear regression models compared to "pure" linear regression (regression without regularization) due to their regularization terms.**

**We clarified this in the revised manuscript.  Rewritten to:**

**"The key variables highlighted by the random forest model are chosen as predictors in Multiple Linear Regression (MLS). MLS, a statistical method employing multiple explanatory variables to forecast the response variable, operates within a linear framework, where the relationship between predictors and response follows the form Y = Xß. However, linear regression techniques can be characterized by complexity and/or overfitting of the model. In this context, Least Absolute Shrinkage and Selection Operator (LASSO) and RIDGE regression emerge as broader forms of linear regression models, incorporating regularization terms, unlike "pure" linear regression lacking such regularization." (~151 – 157)**

o Transformations and distribution of data: the text does not mention the problem of positive skewness (typical of the airborne pollutant concentrations world) (Mudelsee). Clearly, the prediction changes considerably if transformations (e.g., logarithm) are applied to adjust for skewness or if general

models with non-Gaussian distributions (e.g., GLMs and GAMs) are used. Where do you stand with respect to this problem?

**Thank you for the comment. We agree that the air pollution concentration is also not normally distributed in our study and this is indeed a concern. Our study uses the results from Lu, 2023, who used the same dataset as our global dataset. In their study, careful considerations have been given to different transformations, likelihood functions, and loss functions to address the issue of non-Gaussian distribution, with a detailed discussion of the results. It was found that using a transformation, likelihood function, and loss function that matches with the more-likely distribution (the Gamma distribution) does not improve the modelling results but worsened the prediction errors and the uncertainty quantification. In future study we aim at uncovering the reason for these model behaviours.**

**We add this issue in the revised manuscript in discussion.**

**Added:**

**Complementary, airborne pollutant concentrations are often positively skewed. To adjust for positive skewness, transformations can be applied but also cause prediction changes which currently are not revised in our research. Simultaneously, Lu et al. (2023) examine several techniques such as transformations, likelihood functions, and loss functions to address the issue of non-Gaussian distributions. Thereby, they observed that using a transformation, likelihood function, and loss function that matches with the more-likely distribution (i.e. Gamma) does not improve the modeling results but worsened the prediction errors and the uncertainty quantification (Lu et al, 2023). (~428-434)**

o There are dozens of linear models and spatio-temporal mixed-effects models in the literature that provide a fair trade-off between interpretability and predictive ability. In the text, none of them are mentioned. I do not intend to cite specific ones, but just type in Google scholar "spatio-temporal models" to retrieve them. I would suggest starting with the spatio-temporal modeling of Wikle-Cressie (who have made history in this branch of research) and colleagues [1, 2];

**Thank you for the suggestion. We agree that the spatiotemporal modelling works from Wikle Cressie is a great reference and provide inspiring perspectives. We also agree that the spatiotemporal mixed-effect models are making impressive progresses in improving both predictive ability and model interpretability. What is slightly confusing to us regarding the comment is that our study has not reached to the next milestone of spatiotemporal modelling but so far confined into spatial modelling, as many issues remain at this level. We agree that missing the temporal dimension add difficulties in interpretation, uncertainty assessment, and prediction also over space, but joint spatiotemporal modelling greatly complicated the modelling and we believe it is more illustrative and apprehensible to firstly study at a lower dimensionality.**

**We add in the revised manuscript about the future vision of spatiotemporal mixed-effect modelling.**

**Added: "Simultaneously, the absence of the temporal dimension poses challenges in interpretation, uncertainty assessment, and spatial prediction. Still, joint spatio-temporal modeling greatly complicated the modeling and we believe it is more illustrative and reprehensible to firstly study at a lower dimenstionality." (~380)**

o The use of cross-validation is relevant for assessing the predictive ability of models. However, remember that random K-fold, as well as LOOCV, are studied for the crosssectional world. In a spatiotemporal context, they require ad-hoc adjustments that preserve the correlation structure in time and space. In this regard, see the work of Meyer-Pebesma [3-8] on the role of spatial CV and how it is the ideal substitute for random K-fold in the context you study.

**Thank you for this comment. We agree with your comment that the cross-validation methods is highly relevant for assessing the predictive ability of models. We are aware of the literature and critiques regarding spatial cross validation, as well as various cross validation methods. However, we considered spatial cross validation methods not suitable for our study. The reason is well explained in two discussions regarding spatial cross validation methods, Wadoux (2021) and Lu (2023). We agree with the arguments in these two papers consider randomly bootstrapped cross validation suitable to the accuracy assessment of our study.**

**We add this discussion in the revised manuscript.**

**Added: "Wadoux et al. (2021) argue that standard cross-validation (i.e. ignoring autocorrelation) results in smaller bias than spatial cross-validation.400
Moreover, they state that spatial cross-validation methods should not be used for map assessment as they have no theoretical underpinning, while standard cross-validation is applicable and is sufficient in clustered data scenario's (Wadoux et al., 2021; Lu et al, 2023)"**

Also, for the sake of completeness, I suggest adding the word "out-of-sample performances" every time you use CV because it must be clear to reader that all the metrics are computed in a training-test framework to assess predictive capacities of model (and not in-sample fitting);

**Added the words "out of sample performances" before "CV".**
o Feature selection involves an overwhelming number of alternative techniques: the authors use Shapley values, variables importance, penalty (lasso and ridge), bust subset, etc. Ideally, only one method should be chosen to select relevant covariates so that model results are comparable and not data-dependent. Similarly, it is somewhat problematic to have two or more CV schemes (20-fold and LOO) that prevent proper comparison of the models;

**The application of shapely values is used for all global models whereby comparisons between global models are made, not taking into account local models as the approach for the local models is indeed different. This can be attributed to the following. Due to the poor performances by the random forest over all the local station measurements (supplementary, figure 10a, 10b, and 10c), and per spatial group (supplementary, table 2), the best model is not selected by the random forest algorithm and cross-validated Shapley approach. Rather, the best subset regression is used for variable selection.**

o When reading, I had the feeling (probably wrong) that there was a misunderstanding of the statistical tools used. For example, box plots do not assess the "variance" but the "variability" of a phenomenon and use the median (not the mean) as a reference point, as it is robust to the presence of outliers (frequent in air quality).

**Thank you, we changed the terminology to variability and median**

**Specific comments and technical corrections:**

• Abstract, line 2: "model and predict air pollution over space and time"
**No temporal focus**
• Section 2.1, page 5, row 143: the authors state that "We used the precipitation from weather stations (National Centers for Environmental Information, 2017)". Why not directly using Copernicus ECMWF ERA-5 data, which naturally cover the whole Europe with a fine scale (compared to the study area)? Since you used spatial interpolation (ordinary kriging), how do you account for the interpolation uncertainty generated by this approach? How does it reflect on the following stages?

**That is a difficult and good question. ECMWF ERA-5 provides a precipitation at 31 km grid resolution. There are also many other products that model precipitation. We firstly need to know if the precipitation significantly affects the air pollution, that is the reason that we chose to use a reliable source of data and simple interpolation method. It is known that spatial interpolation method such as kriging provide reasonable interpolation for precipitation in Europe (E. Lupikasza 2006).**

**In our study we did not find precipitation to be significant or important. If precipitation contribute significantly to our statistical models, an interesting next-step is indeed comparing different precipitation products.**

**It is also a very good question regarding considering or incorporating the uncertainty of the predictors in the model. It is difficult to account for the uncertainty of each predictors or data source directly, for example by inputting distributions for each data point into the model. However, a probabilistic model could quantify the uncertainty separately from data and model (Hüllermeier and Wägeman 2021; Kendall and Gal 2017). This however deviate the goal of this study.**

**E. Lupikasza, Interpolation methods for precipitation fields in Europe, Geophysical Research Abstracts, Vol. 8, 06493, 2006**

**Hüllermeier, Eyke, and Willem Waegeman. "Aleatoric and epistemic uncertainty in machine learning: An introduction to concepts and methods." *Machine learning* 110.3 (2021): 457-506.**

**Kendall, Alex, and Yarin Gal. "What uncertainties do we need in bayesian deep learning for computer vision?." Advances in neural information processing systems 30 (2017).**

• Section 2.1, formula after row 165: please, explicitly define the symbols/quantities used in the formula. In its current form, it's not easy to understand how the traffic is computed; **Replaced to supplementary; added in main text: "The formula for calculating average hourly traffic can be found in supplementary, equations." (~137).**

• Section 2.2

o Row 171: The average median of what? which values are you considering to rank the features? (later on we discover that you take the median of the rankings… but here it is not clear) **Removed: "The average median of"**

o Rows 175-177: Being the first time you cite LASSO, lightgbm and xgboost models, I suggest using the extended names followed by the acronyms. Also, add some theoretical references on the models (e.g., papers explaining the full methodology); **Full methodology can be better used in supplementary, I assume, as paper (*main text*) is of considerable size already. In main text added: "The equations for the ensemble trees can be found in supplementary, equations." (~149)**

o Row 178: for an extensive comparison in assessing the spatio-temporal prediction accuracy of tree-based methods, linear mixed models and geostatistical mixed models I suggest the papers from the Fassò research group [9-12]; **Again, no focus on the temporal in this paper.**

• Section 2.3

o Actually, the considered models are only described in words but it is difficult to compare it from the analytical perspective. I suggest adding a synoptic table which synthesize the characteristics of the considered models. For instance, the table could state if a model explicitly considers (or not) spatial, temporal or spatio-temporal components (e.g., spatial random effects), if a model is penalized or not (e.g., LASSO), if a model includes covariates or not (e.g., ordinary kriging); **Added table 3 that takes into account model complexity and the consideration of the spatial component**

o Section 2.3.1: can you consider including the recent works on spatial random forests, which extend classical RF to a spatial prediction context [13]? As the aim of the paper is to assess the spatial prediction accuracy of models, this new class could improve a lot your findings;

**Interesting read, I integrated some theory of that paper in the discussion: "Moreover, Patelli et al. (2023) identify three main categories in which random forests can be linked to spatial data, being pre-, in-, and post-processing. While random forest performance is linked with spatial groups in our study (which can arguably be linked to a form of post-processing), there is potential in better integrating spatial data in ensemble tree based models such as random forests, to potentially increase predictive performance (Patelli et al., 2023)" (~398). A further integration of the theories discussed (spatial autocorrelation in random forest modeling) in Patelli into this paper, unfortunately is out of scope. Furthermore, a lot of different approaches to linking spatial data with random forests are discussed, however, most if not all approaches[1] are in it's infancy and not widely adopted measures.**
* * *
[1] RF with SI: Random Forest with Spatial Information; RF with FFS: Random Forest with Forward Feature Selection; RF-RK: Random Forest Residual Kriging; RF-sGs: Random Forest sequential Gaussian simulation; RF-RK with SI: Random Forest Residual Kriging with Spatial Information; RF-RK with SB: Random Forest Residual Kriging with Spatial Bootstrap; RF-GLS-RK: Random Forest based on GLS Residual Kriging.

o Section 2.3.2, row 214 ($\alpha$): Please, explicitly define the parameter alpha. Also, if alpha refers to the elastic net mixing parameter, with $0 \leq \alpha \leq 1$, then you are considering the elastic net penalization, which is a combination of LASSO and ridge, and not exactly LASSO or ridge;

**Thank you for you acknowledging this. The text now specifically mentions that the alpha refers to the separate LASSO and RIDGE models.**
• Section 2.4
o Row 224: "N describes a set of n features" is not clear. which is the difference between N and n? What do you mean by payout? Is it the prediction with the N features? Is S the cardinality of the subset of N? **Changed in supplementary, equations**
o Rows 228-229: please, consider rephrasing the whole sentence: the current sentence seems to state that in general/typically Shapley values are embedded in the two alternative CV approaches. However, this seems to be one of your proposals;

**We do not mention mean approach anymore to avoid confusion; figure of mean cv is still apparent in supplementary material.**
o Rows 239-242: Please, consider rephrasing the whole sentence as currently it is confused. My interpretation of Figure 2 is that a sensible/remarkable prediction accuracy improvement is obtained when considering at least 12 predictors. However, the improvement is marginal considering more than 12 covariates (the curves become flat);

**Added:**

**A remarkable prediction accuracy improvement is obtained when considering at least 12 predictors. However, the improvement is marginal considering more than 12 covariates as the curve flattens. (~192)**

o Section 2.4.2 (best subset regression): usually, best subset is used in a linear regression framework. Still, it is not clear to me if you are considering its application in linear or non-linear (in this case, which model?) models. Also, best subset regression is typically affected by computational inefficiency as it requires the computation of 2^k-1 models (where k=30 in the number of covariates). Do you have any insights about the computational burden of this step?

**Added: "Rather, the best subset regression is used for variable selection for the local models" (~197)**

**Computationally unfeasible to continue with a higher k number; moreover global models indicate that the prediction accuracy of a model seems to stagnate way before 30, taking this as an inspiration to set a limit.**

o Section 2.4.2 (linear models): still, it is not clear to me which class of mixed-effects models are you considering. Please, can you state (in Appendix or Supplementary Materials) the exact formulae and parameter specifications (i.e., which is the structure of the random effects? Are they i.i.d. sequence of Gaussian RVs or are spatio-temporally structured?). Also, later on you state that the "... linear models (i.e., LASSO and ridge) ...": why not considering a multiple linear regression without penalization? This last model should be directly comparable with penalized approaches.

**Equation for mixed effects model can now be found in supplementary, equations; parameter specifications in supplementary, parameters. For the local model, a multiple linear regression without penalization is indeed considered.**

• Figure 3 (caption): I would say that the upper and lower whiskers provide information about the overall variability of the estimates rather than "variance". Box-whisker plots are typically computed using the IRQ-rule, that is, the whiskers are +-1.5 x IQR (interquartile range, i.e., X_0.75 - X_0.25). Also, box-plots typically use the median as central value. Is the orange line the median? If so, why do you talk about "mean statistic" in the first paragraph of Section 3.1.1? In air quality statistics there is a huge difference among robust (median) and nonrobust (mean) methods for assessing the centrality of air quality distributions.

**Thank you, we changed the terminology to variability and median**

• Section 3.1.1
o row 286: what do you mean by "uncommon"?

**Changed to "may be present" (~285)**

o right before Table 2: Spatial characteristics is a fundamental feature in air quality statistical modeling. Indeed, local air quality is substantially affected by local weather and environmental conditions. However, why did you not include such variable in the features selection stage? You should be sure about the effective predictive capacity of such variables before including it "a priori". Also, I suppose you used the information through a set of dummy variables (I guess 2 vars). Is that correct? Which one did you choose as reference category? Otherwise, did you used separated/independent models by category (i.e., you estimated all the previous models only for Urban and then for low pop and then for far from road)? In the latter case, you should compare the results with the full dataset very carefully as in the submodels you are ignoring a large part of the information contained in the full data;

**Local weather and environmental conditions (e.g. wind, temperature, precipitation) are considered in the feature selection stage. I separated the models by spatial group (urban, lowpop, far from road) on which table 2 (now 3) is based. The table shows some NO2 descriptives per spatial group and already unravels some significant differences in NO2 variability.**

o row 302: please, clearly state the definition of "more discrete" outcomes or models;

**We removed this already, based on 1[st] reviewer**
o rows 307 on: whenever you cite a specific place (e.g., Harleem), please make sure that the area is recognizable on the maps. Where is Harleem in Figures 4 and 5? The same comment holds for all the other cities/locations. **Added a spatial reference in supplementary figure (figure 10d - spatial references Amsterdam area)**
• Figures 4 and 5: they compare different models for different locations. Can you justify this choice? It seems an unfair comparison: to understand the effects of models one should compare different models at the same locations. The comparison you propose is meaningful only if you are sure that, independently on the local conditions, the predictions are comparable (thus there is no spatial effect) and the only relevant factor is the model's definition;

**Not entirely true I would say, figure 4 projects the predictions of the global models on the same spatial extent. Figure 5, a comparison between a linear (RIDGE) and non-linear (random forest) is made for every spatial extent, being Bayreuth, Hamburg, Utrecht (in supplementary, again each global model has a prediction for each spatial extent, making comparisons between global models for a spatial extent possible).**

**Moreover, based on figure 5, you still can say something about the influence of every predictor on the prediction patterns (no2) for every spatial extent.**

• Figure 6: still, if you use box-plots, then the central value you are comparing is the median. Also, it is not clear to me what you are representing on the box-plots. Are they the distribution of the estimated NO2 concentrations at every point (if so, how many points did you interpolate?) in a specific area (Figures 4 and 5) or are they temporal predictions at some locations (in this case, which locations?) or are the spatio-temporal predictions? Also, where are the results associated with linear mixed models?

**I removed this figure out of the main text so, NA**

• Section 3.1.2: why did you move from a 20-fold CV for global model assessing to a N-fold (LOO) CV for local models? This choice introduces some issues when comparing models as the predictions are computed using sample sizes;

**Because the local dataset has fewer data, therefore a LOO approach suits better.**

I removed a part of 3.1.2 and moved it to the methodology where it is more suitable. Concerns: **"With the mixed-effects model, fixed and random effects are included. Fixed effects consist of the most influential predictors while random effects account for potential spatial trends in the data. The spatial trends in the data related to observations being clustered in a way. The spatial character of the observation, i.e. whether an observation is situated in an urban area, low-populated area, or far from road area, accounts for the random effect in the model. In contrast, the linear model composes all the fixed effects while neglecting the possibility of observation clustering. Additionally, two kriging methods are used for local modeling, being ordinary- and universal kriging."**

• Table 4: Where are the machine/statistical learning results (i.e., lightgbm, xgboost, random forest)? What is "linear model" and which is its relationship with ridge and LASSO? Why do you compare a different set of models? As for the mapping, if different models are used to compare local and global modeling, the comparison will be biased and unfair;

**Using different models is not worrisome perse, but using different data is hence we divide between global and local and compare the models within the groups. Table 4 only applies to the local models (as it is part of the "local model" section); since the machine/statistical learning results are based on the global data, the machine/statistical learning are irrelevant for table 4.**

• Figure 9: are Kerckhoffs's data used to train the models? Why not using the actual NO2 observations (used as response variable of the models) as benchmark? Also, I would plot the original NO2 concentrations used as response variables in the models. They are the actual benchmark. **No, our data originates from the municipality of Amsterdam (open source) while Kerckhoffs et al (2019) use other sources.**

**Actions by author(s)**

**Added:**

- Table 1. Descriptive statistics for each relevant variable in the determination of spatial groups for the local- and global datasets
- Added table 2 and 3 about model specifications.
- "The formula for calculating average hourly traffic can be found in supplementary, equations." (~137).
- "The equations for the ensemble trees can be found in supplementary, equations." (~149)
- "The key variables highlighted by the random forest model are chosen as predictors in Multiple Linear Regression (MLS). MLS, a statistical method employing multiple explanatory variables to forecast the response variable, operates within a linear framework, where the relationship between predictors and response follows the form $Y = X\beta$. However, linear regression techniques can be characterized by complexity and/or overfitting of the model. In this context, Least Absolute Shrinkage and Selection Operator (LASSO) and RIDGE regression emerge as broader forms of linear regression models, incorporating regularization terms, unlike "pure" linear regression lacking such regularization." (~151 – 157)
- "The relevant equations can be found in supplementary, equations." (~172)
- A remarkable prediction accuracy improvement is obtained when considering at least 12 predictors. However, the improvement is marginal considering more than 12 covariates as the curve flattens. (~192)
- Added: "Rather, the best subset regression is used for variable selection for the local models" (~197)
- ""Moreover, Patelli et al. (2023) identify three main categories in which random forests can be linked to spatial data, being pre-, in-, and post-processing. While random
forest performance is linked with spatial groups in our study (which can arguably be

linked to a form of post-processing), there is potential in better integrating spatial data in ensemble tree based models such as random forests, to potentially increase predictive performance (Patelli et al., 2023)" (~398).

- Added: "Simultaneously, the absence of the temporal dimension poses challenges in interpretation, uncertainty assessment, and spatial prediction. Still, joint spatio-temporal modeling greatly complicated the modeling and we believe it is more illustrative and reprehensible to firstly study at a lower dimensionality." (~380)

- Added: "Wadoux et al. (2021) argue that standard cross-validation (i.e. ignoring autocorrelation) results in smaller bias than spatial cross-validation.400 Moreover, they state that spatial cross-validation methods should not be used for map assessment as they have no theoretical underpinning, while standard cross-validation is applicable and is sufficient in clustered data scenario's (Wadoux et al., 2021; Lu et al., 2023)"

- "Complementary, airborne pollutant concentrations are often positively skewed. To adjust for positive skewness, transformations can be applied but also cause prediction changes which currently are not revised in our research. Simultaneously, Lu et al. (2023) examine several techniques such as transformations, likelihood functions, and loss functions to address the issue of non-Gaussion distributions. Thereby, they observed that using a transformation, likelihood function, and loss function that matches with the more-likely distribution (i.e. Gamma) does not improve the modeling results but worsened the prediction errors and the uncertainty quantification (Lu et al, 2023)." (~428-434)

- Supplementary equations – equations relating to several elements that are discussed in the main text: traffic volume; ensemble trees; lasso- and ridge regression; kriging methods; mixed effects model; feature selection

- Supplementary parameters – mainly supportive to methodology section

- Supplementary, spatial reference (e.g. Haarlem)

**Key removals**

- Equation(s) (1)(2)
- Table 1 (data -descriptives)
- Figure 6 (Distribution predicted NO2 (µg/m3) per model and per location) (now part of supplementary figures)
- Figure 8 (Distribution predicted NO2 (µg/m3) per model (local) with outliers correction. LR = linear regression, LRsp = linear regression accounting for spatial groups, MEM = mixed- effects model, UK = universal kriging, UKsp = universal kriging accounting for spatial groups, OK = ordinary kriging) (now part of supplementary figures)

---

## Referee Report (RR1)

Thanks for your substantial work. I think this is a good work to stimulate discussion on with predictive mapping of environmental factors. I would like to recommend a major revision to address the following comments.

**Major:**

Line 75-80.

- Could you confirm if the global dataset overlaps with the local dataset? Even though they are from different sources, some of them might be the same stations but with different names.

Line 100-105.

- Consider renaming the three groups. The current version reads a bit misleading as it looks like they partially overlap.

Line 110-115.

- If the definition of the three groups changed between local and global models, would it still be fair to compare their model performance? For example, the threshold adjusted from 0.75 to 0.5 for the "urban" in the global and local models. It could impact the conclusion. In the result section, the different distributions of model predictions could stem not only from the levels (global or local) but also from the different definitions of the "urban" group.
- Have you tried to develop models trained with the balanced numbers of instances across spatial groups? i.e., the same number of instances in each group, as the statistic learning model can be easily biased due to the unbalanced distribution of training instances in each category.

Figure 6.

- Comparing the spatial variations of predictions between global and local models is challenging due to the differing algorithms used.

**Minor:**

Abstract:

1. What is your final conclusion or the key message? Better to specify it in the final sentence in the abstract.

Line 75-80.

-Please clarify the spatiotemporal resolution of the models.

- Add the number of stations of the two datasets.

Line 180-185.

- Please specify why the feature selection. If it is about avoiding collinearity, why not use the VIF value?

Line 190.

- What do you mean by the out-of-sample cross-validation? Did you use external/thirdparty data sets (other than the global/local measurements)?

Line 225-230.
- 20-fold cross-validation means the training set is divided into 20 parts. The global model was trained with 482 observations. In each iteration, only 24 observations are replaced. It is fine to do 20-fold cross-validation. But for the small size of samples, it is not a proper choice.

Line 235-240
- This part belongs to the discussion section.

Figure 6.
- Would it be possible to plot also the distribution of mobile predictions from Kerckhoffs? Another crucial aspect of the air pollution map is the spatial variations. I expect to see the different levels of variations captured by global and local models as well as fixed-site vs mobile measurements.

Line 290.
- UK = universal kriging? Please use the full name in the bracket.
- What is the linear model in the table? Lasso? Ridge? Why are the algorithms used for global and local models not aligned?

Line 325-330.
- It is great to involve external datasets for cross-checking.
- What is the true reason for filtering out outliers? Enhancing the low correlation is not a proper reason. Rephrase, please.
- Use the past tense. This is something you have done.

Line 335.
- Another reason for kriging is its stationary assumption.

Line 360-375.
- If the temporal analysis is not performed, please put it into the limitations or future work section.

Figure 7.
- Can you specify which models are global and which are local models directly in the figure? To increase the readability.

Line 380-405.
- Another nice paper I found that discusses also the difference between the global and local models is "Integrating large-scale stationary and local mobile measurements to estimate hyperlocal long-term air pollution using transfer learning methods". They found also a significant improvement in modeling performance in urban background

areas when involving global knowledge. This would be a good citation.

Line 445-450.
- Need to also mention the missing meteorological information.

---

## Referee Report (RR2)

**General comments:**

The authors didn't provide sufficient feedback to the comments of reviewer 2 in most cases. I have summarised below some of the specific comments that deserve adequate feedback. In addition, the authors need to answer some theoretical questions underpinning air pollution exposure modeling;

1. Air pollutants are not only defined spatially but temporarily and both characteristics interact in a complex relationship that can't be disentangled especially with the approach used by the authors. They also use just one pollutant (NO2) which is very localized. This might explain the performance difference reported between global and local models.
2. The assumption behind using traffic and population-related variables to construct their models is not rooted in the literature because other land-use and temporal variables have been reported to explain variability in air pollution levels.
3. The statistical approaches used for this analysis are not contextualized in their manuscript and the supplementary materials provided are too theoretical.
4. I doubt this paper will benefit from any other revision because the issues are related to how the research question was conceptualized. Does this work have any potential to add anything new to the existing literature in this field?. My answer is NO.

**Specific comments**

R2 comment: The overall readability of the text is very poor: the presentation of machine/statistical learning models is very superficial (there are no formulas and no technical modeling aspects are discussed); To make the paper more readable, the formulas are added in the supplementary material.

Author's Feedback: **A reference to the equations and technical aspects of the considered models is now available in supplementary, equations.**

My comment: Although, the authors have added a document with the general equation of the models considered in their analysis. However, these equations are mostly general equation description found in textbooks or papers. The application of these model parameters to their data wasn't discussed in detail. I would have expected the authors to include some equations summarising their models in the manuscript to complement the text provided.

R2 comment: The paper is very long and confusing: reading requires continuous jumping from one section to another to understand what models and assumptions the authors are analyzing.

Author's Feedback: I changed the structure of the methodology. In the renewed methodology, the data (2.1) is introduced first, followed by a short elaboration per model used in the next subchapter (2.2); subsection 2.3 elaborates on feature selection which is an important part as it determines the relevant variables for the modeling; the last section of the methodology provides insights into how the models are evaluated and used, thereby showing the relevant models in a table overview.

**My comment: In general, the paper is still confusing. I struggle to understand why the decisions to construct these models. What other information do these different models add to understanding the spatial pattern in N02 exposure?**

R2 comment: Methodology: Models are presented without specifying their technical characteristics, differences and rationale for their use. No formulas explaining the structure of the models (e.g., the spatiotemporal structure of random effects) are included by the authors. A paper using statistical methodology should never assume that the reader is aware of the methods;

Author's Feedback: Information on technical characteristics, differences, and rationales for modeling can now be found in supplementary equations and supplementary parameters. The temporal aspect is neglected in the models unfortunately, as this is outside the scope of this research, however, should be addressed in future research.

My comment: The technical characteristics discussed in this section are very theoretical and largely didn't describe why and how these methods were applied to this analysis.

R2 comment: There are dozens of linear models and spatio-temporal mixed-effects models in the literature that provides a fair trade-off between interpretability and predictive ability. In the text, none of them are mentioned. I do not intend to cite specific ones, but just type in Google scholar "spatio-temporal models" to retrieve them. I would suggest starting with the spatio-temporal modeling of Wikle-Cressie (who have made history in this branch of research) and colleagues [1, 2];

Author's Feedback: Thank you for the suggestion. We agree that the spatiotemporal modelling works from Wikle Cressie is a great reference and provide inspiring perspectives. We also agree that the spatiotemporal mixed-effect models are making impressive progresses in improving both predictive ability and model interpretability. What is slightly confusing to us regarding the comment is that our study has not reached to the next milestone of spatiotemporal modelling but so far confined into spatial modelling, as many issues remain at this level. We agree that missing the temporal dimension add difficulties in interpretation, uncertainty assessment, and prediction also over space, but joint spatiotemporal modelling greatly complicated the modelling and we believe it is more illustrative and apprehensible to firstly study at a lower dimensionality. We add in the revised manuscript about the future vision of spatiotemporal mixed-effect modelling. Added: "Simultaneously, the absence of the temporal dimension poses challenges in interpretation, uncertainty assessment, and spatial prediction. Still, joint spatio temporal modeling greatly complicated the modeling and we believe it is more illustrative and reprehensible to firstly study at a lower dimenstionality."

**My comment:** This is a fundamental issue for a paper titled: **"A close look at using national ground stations for the statistical modeling of NO2"**. I wonder about the gap this paper is trying to fill. We know already that air pollutants are defined by their spatial and temporal characteristics. Also, the interaction between these characteristics can be complex – more reason why complex models that can account for these interactions are becoming more popular in air pollution exposure science. So a statistical modeling of NO2 at local and national levels without accounting for temporality would not tell us the full story of these complexities in air pollution exposure modeling. The title also didn't make this explicit.

R2 comment: Section 3.1.2: why did you move from a 20-fold CV for global model assessing to a N-fold (LOO) CV for local models? This choice introduces some issues when comparing models as the predictions are computed using sample sizes;

Author's Feedback: Because the local dataset has fewer data, therefore a LOO approach suits better. I removed a part of 3.1.2 and moved it to the methodology where it is more suitable. Concerns: "With the mixed-effects model, fixed and random effects are included. Fixed effects consist of the most influential predictors while random effects account for potential spatial trends in the data. The spatial trends in the data related to observations being clustered in a way. The spatial character of the observation, i.e. whether an observation is situated in an urban area, low-populated area, or far from

road area, accounts for the random effect in the model. In contrast, the linear model composes all the fixed effects while neglecting the possibility of observation clustering. Additionally, two kriging methods are used for local modeling, being ordinary- and universal kriging.

My comment: The authors argued in their previous response to the comment about spatial cross-validation that spatial cross-validation is not a suitable measure of model performance because they have no theoretical underpinning. I am confused as to why the leave one out cross-validation – a spatial cross-validation method was then used to assess the local models' performance.  It's also surprising that the authors compared the global model performance assessed using random cross-validation to local models performance assessed using spatial cross-validation. What is the theoretical underpinning for this ?. This is not an appropriate way to go about it and the conclusion from this comparison is questionable.

---

## Referee Report (RR3)

This study addresses the challenge of accurately modeling nitrogen dioxide (NO2) concentrations, which is essential for understanding air pollution's health and environmental impacts. Given that NO2 levels vary significantly across different spatial settings, especially in urban areas, the authors investigate how various statistical and machine learning models perform across urban, suburban, and rural regions. By comparing global and local models—trained on datasets from Germany, the Netherlands, and specifically Amsterdam—the study evaluates the strengths and limitations of each model type.

The authors tackle this problem by creating spatially-defined groups based on traffic volume and population density, which allows the models to account for spatial heterogeneity and examine prediction patterns across different zones. They apply both linear and non-linear models, as well as mixed-effects and kriging methods, to see how well each approach handles the spatial intricacies of NO2 concentrations. Model performance is evaluated through standard metrics like $R^2$, RMSE, and MAE, revealing that ensemble-based models perform best in rural areas, while urban areas remain challenging due to data heterogeneity. This methodology highlights the critical role of spatial grouping and suggests that relying solely on prediction accuracy without considering spatial context may lead to misleading results.

The study is interesting and investigates an important aspect of using data-driven methods to predict air pollutants on a large scale. There are however some concerns I have identified before the study can be published, listed below.

**Methodology**

1. (section 2.1 Data) The authors refer to the two datasets as global and local. However, the global dataset only contains data originating from two neighboring countries, with similar characteristics. I suggest to change the name from global to something more appropriate as these two countries do not reflect the global stage.
2. Line 100-105: Can the authors discuss the inclusion of the distance to roads feature in clustering the regions or include a citation to a study that supports such distinction? Traditionally, the classification of urban, sub-urban and rural areas is based on population alone.
3. Line 111-103: Discuss this more, as it seems like these two statements are contradictory.
4. Line 135: 50000 estimators seems extremely excessive for the boosting algorithms. Traditionally, the number of estimators is in the hundreds. I would suggest the authors to discuss the reasoning behind this. While gradient boosting algorithms are resilient to overfit, they are not overfit-proof and including a very large number of estimators has the potential for overfitting.
5. Line 138, 141: Specific references to the supplementary material is missing. e.g. In which table are these results presented?
6. Line 150: SI table 4 shows the results for Linear, LASSO and Ridge and SI Table 5 shows the results for Random Forest, LightGBM etc.
7. Section 2.2.2 and 2.2.3: The description of the methodology for the modeling part is superficial. The authors should expand these sections significantly, especially section 2.2.3 which describes the mixed-effects model and the Kriging method.
8. Section 2.3: The SHAP values plot should be in the main text of the manuscript instead of the supplementary material, as it contains useful information that are used in the main study.

9. Section 2.3: It's not clear whether the authors have normalised the data used here for the machine learning algorithms. From the SHAP figure it seems that the target variable range is not normalized. Normalization of the input and target variables to the 0 to 1 range ensures that all the input variables are equally weighted (unless the setup of the model specifically requires asymmetric weights) and the input variables with the largest range (or absolute values) do not dominate the others.
10. Section 2.3: Consider expanding this section as it's not clear which predictors are selected and how the process is implemented. Also, a table of all the predictors used and their origin, range and name could be useful to the reader.
11. Line 192: In Kerckhoffs et. al (2019) the maps were produced by measuring for limited time periods using mobile sensors. The temporal resolution of the predictions of the models here deal with much coarser temporal resolutions.

**Results**

1. Line 201: How was this 20-fold validation performed? Since the number of data points is small, I suggest to perform cross-validation with a small number of folds (e.g. 5). How many data points were selected in each fold and on which set these metrics were evaluated on?
2. A graph of ground truth vs predictions would be beneficial here to identify edge cases at which the algorithms do not perform well.
3. Line 216: It's not clear what the spatial resolution of the predictions is nor how these maps were created.
4. Table 6: Another useful metric that could be used to gauge the significance of the RMSE and MAE metrics would be percentage error wrt to ground truth.
5. Line 256: When was leave-one-out cross-validation used before this? Discuss how this was implemented.
6. Line 260: A significant limitation of the study setup is the fact that the most heterogeneous group (urban) is the least represented in terms of number of data points. This should be discussed by the authors.
7. The large number of models and the use of different set of models for each group makes these comparisons very difficult. Also, why have the authors used similar models (e.g. LightGBM and XGBoost) which makes the comparisons even more difficult to follow.
8. Line 290: "These outliers were removed..." Discuss this choice more. How many points were removed and why do you think these points performed poorly?

**Disussion**

1. Line 323 "While a well-trained model..." Not clear what this sentence conveys to the reader.
2. LIne 334: I would argue that spatial cross-validation is essential in this kind of models, as it ensures that the model learns sufficient representations to generalise to other regions that do not have any ground stations. In this case, spatial cross-validation would be beneficial within the groups selected. i.e. ensure that the model generalises well within the urban cluster
3. Line 342-346: This is counter-intuitive. It pollutes the urban group by including areas with less population that the initial definition (upper 75% quartile) but it was necessary to expand the size of the dataset to a sufficient level. The authors should make this clear and

address it as a limitation of the study. While it's understandable this was a necessary step in the experimental setup of the study, it needs to be clearly addressed.

**Conclusions**

1. "In this study, we understand..." Consider changing the word understand to investigate

---

## Author Response (AR2)

**Again, we appreciate your time and effort into the feedback given. Our replies to your feedback is expressed in bold font.**

**Feedback 1**

Thanks for your substantial work. I think this is a good work to stimulate discussion on with predictive mapping of environmental factors. I would like to recommend a major revision to address the following comments.

**There is now a pdf including all the supplementary figures.**

Line 75-80. - Could you confirm if the global dataset overlaps with the local dataset? Even though they are from different sources, some of them might be the same stations but with different names.

**There is no overlap. Also see figure 4 and figure 5 of the supplementary figures.**

Line 100-105. - Consider renaming the three groups. The current version reads a bit misleading as it looks like they partially overlap.

**The three groups have been (partly) renamed to" urban", "suburban", and "rural" spatial groups so that there is no indication that they might overlap.**

Line 110-115.
 - If the definition of the three groups changed between local and global models, would it still be fair to compare their model performance? For example, the threshold adjusted from 0.75 to 0.5 for the "urban" in the global and local models. It could impact the conclusion. In the result section, the different distributions of model predictions could stem not only from the levels (global or local) but also from the different definitions of the "urban" group.

**The adjustment of the threshold from 0.75 to 0.5 for defining "urban" areas in the local dataset addresses the inherent differences between global and local datasets. The local dataset has fewer samples but features higher population densities. In this context, a lower threshold for "urban" classification is necessary to accurately capture areas with the highest population concentrations. By lowering the threshold, we can better represent the spatial distribution of densely populated areas in the local context, leading to more precise and relevant modeling results.**

**This adjustment allows for a more flexible and context-sensitive definition that better reflects the unique characteristics of the local dataset. However, it's important to approach comparisons between models using different thresholds with caution. Adjusting the threshold is a methodological choice intended to improve the accuracy and relevance of the model in capturing local phenomena.**

**Your point is valid, and I have clarified this rationale in the main text.**

Line 110-115.
- Have you tried to develop models trained with the balanced numbers of instances across spatial groups? i.e., the same number of instances in each group, as the statistic learning model can be easily biased due to the unbalanced distribution of training instances in each category.

**Not necessarily, although adjusting the urban threshold for the local dataset helps to address this issue indirectly. For the global dataset, this is less relevant due to its larger sample size.**

**For the local dataset, 56 observations are classified as "urban," 46 as "suburban," and 30 as "rural." This adjustment was made to partially address the imbalance in the number of instances across these spatial groups, aiming for a more equitable distribution between "urban," "suburban," and "rural" categories. While this doesn't completely eliminate the imbalance, it does help the local model better reflect the higher population density characteristic of urban areas in the local context. We acknowledge that the unequal distribution of instances across groups could introduce bias in statistical learning models, but this threshold adjustment was an initial step to mitigate such effects. This has also been briefly noted in the main text.**

**The decision to adjust the threshold for the local dataset was justified by the need to achieve a fairer distribution of instances across groups:**

- **With a 0.75 threshold, there are 25 samples in the urban group, 56 in the suburban group, and 51 in the rural group.**

- **With the 0.5 threshold, there are 56 samples in the urban group, 46 in the suburban group, and 30 in the rural group.**

Figure 6. - Comparing the spatial variations of predictions between global and local models is challenging due to the differing algorithms used.

**Thank you for the feedback. We acknowledge that comparing spatial variations between global and local models is challenging due to the differing algorithms used. In the revised version of the paper, we have addressed this issue by highlighting how the distinct approaches of the models impact the observed spatial patterns. Figure 6 now explicitly discusses these algorithmic differences and their effects on the comparability of predictions.**

**Minor:**

Abstract: 1. What is your final conclusion or the key message? Better to specify it in the final sentence in the abstract.

**Extended the last part of the abstract.**

Line 75-80.
- Please clarify the spatiotemporal resolution of the models.

**Global Model:**

- **Total Area (km²): 398,087.4**

- **Total Points: 482**

- **Point Density (points per km²): 0.0012**

**Local Model:**

- **Total Area (km²): 196.4**

- **Total Points: 116**

- **Point Density (points per km²): 0.591**

**I've included this information in the main text.**

 - Add the number of stations of the two datasets.

 **Added**

Line 180-185. - Please specify why the feature selection. If it is about avoiding collinearity, why not use the VIF value?

**The feature selection process utilizes Shapley values primarily to identify and prioritize predictor variables with the most significant influence on NO2 concentration levels, enhancing model performance and interpretability. Shapley values are advantageous because they provide a nuanced assessment of each feature's contribution by considering all possible combinations of features. This approach allows for a detailed evaluation of feature importance, accounting for interactions and correlations between features.**

**While the Variance Inflation Factor (VIF) is effective for detecting multicollinearity by measuring how much the variance of a regression coefficient is inflated due to collinear predictors, it does not directly address feature importance or interactions. VIF is primarily a tool for identifying redundant features rather than assessing their contribution to the target variable. In contrast, Shapley values offer a comprehensive measure of how each feature impacts the prediction, including the effect of feature interactions, which is crucial for understanding the model's behavior and improving its performance.**

**Therefore, Shapley values are chosen over VIF because they provide a more holistic view of feature importance and interactions, which aligns with the goal of enhancing model performance and interpretability in this context.**

**The VIF analysis shows that the population variables are correlated with each other, as evidenced by their high VIF values. Despite this, both population_1000 and population_3000 might be crucial predictors of NO2 levels based on the Shapley**

values, indicating they provide significant and complementary information to the model.

**To address the potential multicollinearity while retaining the valuable information from these features, regularization techniques such as Ridge or Lasso regression are employed. These methods can help manage the redundancy while still incorporating the features' predictive power. This is addressed in section 2.2.2 Multiple linear regression. The VIF scores for both the global and local datasets are included in supplementary table 2 and table 3 respectively.**

Line 190. - What do you mean by the out-of-sample cross-validation? Did you use external/third- party data sets (other than the global/local measurements)?

**This out-of-sample cross-validation terminology is implemented since the previous feedback round, as feedback from another reviewer entails:**

**"Also, for the sake of completeness, I suggest adding the word "out-of-sample performances" every time you use CV because it must be clear to reader that all the metrics are computed in a training-test framework to assess predictive capacities of model (and not in-sample fitting);"**

Line 225-230. - 20-fold cross-validation means the training set is divided into 20 parts. The global model was trained with 482 observations. In each iteration, only 24 observations are replaced. It is fine to do 20-fold cross-validation. But for the small size of samples, it is not a proper choice.

**To clarify, we used repeated random sampling validation rather than traditional 20-fold cross-validation. In our approach, we sampled 25% of the observations as test sets in each of the 20 iterations, with the remaining data used for training. This method involved repeated sampling with different random states to ensure robust evaluation. We recognize the limitations of using cross-validation with smaller datasets and appreciate your input on this matter. The relevant part in the main text is adjusted to this.**

Line 235-240 - This part belongs to the discussion section.

**Inserted this part under section "Accounting for spatial groups" in the discussion section**

Figure 6. - Would it be possible to plot also the distribution of mobile predictions from Kerckhoffs? Another crucial aspect of the air pollution map is the spatial variations. I expect to see the different levels of variations captured by global and local models as well as fixed-site vs mobile measurements.

**A new table (8) showing the residuals per model, by comparing it with the open NO2 dataset (Kerckhoffs), is added to the main text. The spatial residuals per global and local model can be found in supplementary figure 22 and figure 23 respectively.**

Line 290.
- UK = universal kriging? Please use the full name in the bracket.
**Adjusted.**
- What is the linear model in the table? Lasso? Ridge? Why are the algorithms used for global and local models not aligned?

**The performances of the algorithms used for global models, perform substantially poorer on the local dataset, also for the Lasso and Ridge algorithms. See the leave-one-out cross validation results (also included in supplementary, table 4; a reference is made in the main. text):**

Line 325-330.
- It is great to involve external datasets for cross-checking.

**Also added a table (8) with model residuals (comparing with NO2 mobile map of Kerckhoffs)**

- What is the true reason for filtering out outliers? Enhancing the low correlation is not a proper reason. Rephrase, please.
 - Use the past tense. This is something you have done.

**Adjusted. To improve the clarity of the correlations between the models and the open NO2 dataset, we addressed some extreme prediction values. These outliers were removed to prevent them from skewing the analysis and to provide a more accurate representation of the correlations. This is mentioned in the main text too.**

Line 335. - Another reason for kriging is its stationary assumption.

**Added**

Line 360-375. - If the temporal analysis is not performed, please put it into the limitations or future work section.

**Shortened the discussion section. The feedback to which this apply, is removed to make the paper more compact.**

Figure 7. - Can you specify which models are global and which are local models directly in the figure? To increase the readability.

**Added in figure 7.**

Line 380-405. - Another nice paper I found that discusses also the difference between the global and local models is "Integrating large-scale stationary and local mobile measurements to estimate hyperlocal long-term air pollution using transfer learning methods". They found also a significant improvement in modeling performance in urban background areas when involving global knowledge. This would be a good citation.

**Thank you for this suggestion. I put it in the discussion (section "Global and local predictions").**

Line 445-450. - Need to also mention the missing meteorological information.

**Shortened the discussion section. The feedback to which this apply, is removed to make the paper more compact.**

---

## Author Response (AR3)

Public justification (visible to the public if the article is accepted and published):
Dear authors

Thank you for revising the manuscript. Based on the latest reviews, a further revision is necessary. Please take this opportunity to clarify what the statistics in Table 1 summarise and the notation of the predictors in Table 2. In this context, please also provide the resolutions of all data sets used.

**Thanks for your suggestions. We added two tables to address this: table 2 and table 3 that show the most important predictors with their descriptive statistics.**

**Review 1**

This study addresses the challenge of accurately modeling nitrogen dioxide (NO2) concentrations, which is essential for understanding air pollution's health and environmental impacts. Given that NO2 levels vary significantly across different spatial settings, especially in urban areas, the authors investigate how various statistical and machine learning models perform across urban, suburban, and rural regions. By comparing global and local models—trained on datasets from Germany, the Netherlands, and specifically Amsterdam—the study evaluates the strengths and limitations of each model type.

The authors tackle this problem by creating spatially-defined groups based on traffic volume and population density, which allows the models to account for spatial heterogeneity and examine prediction patterns across different zones. They apply both linear and non-linear models, as well as mixed-effects and kriging methods, to see how well each approach handles the spatial intricacies of NO2 concentrations. Model performance is evaluated through standard metrics like R², RMSE, and MAE, revealing that ensemble-based models perform best in rural areas, while urban areas remain challenging due to data heterogeneity. This methodology highlights the critical role of spatial grouping and suggests that relying solely on prediction accuracy without considering spatial context may lead to misleading results.

The study is interesting and investigates an important aspect of using data-driven methods to predict air pollutants on a large scale. There are however some concerns I have identified before the study can be published, listed below.

**Methodology**

1. (section 2.1 Data) The authors refer to the two datasets as global and local. However, the global dataset only contains data originating from two neighboring countries, with similar characteristics. I suggest to change the name from global to something more appropriate as these two countries do not reflect the global stage.

**While we acknowledge that the "global" dataset includes data from only two neighboring countries with similar characteristics, our choice of terminology was intended to distinguish between the larger-scale, cross-border dataset (encompassing cross countries) and the smaller, localized dataset. In this context, "global" is used relatively, to denote a broader scope compared to the "local" dataset.**

**Changing the terminology to "country" versus "multi-country"/"cross-country" might introduce further ambiguity, as it would still not fully capture the comparative distinction we aim to convey. To address the concern raised, we have clarified the relative nature of these terms in the text to ensure readers understand our intended framing (section 2.1).**

2. Line 100-105: Can the authors discuss the inclusion of the distance to roads feature in clustering the regions or include a citation to a study that supports such distinction? Traditionally, the classification of urban, sub-urban and rural areas is based on population alone.

**$NO_2$ is well-documented as a traffic-related pollutant, with concentrations often strongly influenced by proximity to major roads and traffic density. Therefore, incorporating distance to roads as a feature in the spatial group definition aligns with the pollutant's source characteristics and its spatial distribution patterns. While traditional classifications of urban, suburban, and rural areas often rely solely on population data, the inclusion of road-related variables provides additional relevant distinction in the context of air pollution studies (Chen et al., 2019. Lu et al., 2023).**

**Chen, J., de Hoogh, K., Gulliver, J., Hoffmann, B., Hertel, O., Ketzel, M., ... & Hoek, G. (2019). A comparison of linear regression, regularization, and machine learning algorithms to develop Europe-wide spatial models of fine particles and nitrogen dioxide. *Environment international*, *130*, 104934.**

**Lu, M., Cavieres, J., Moraga, P. (2023). A Comparison of Spatial and Nonspatial Methods in Statistical Modeling of NO. *Geographical Analysis*, *55*, 703-727.**

3. Line 111-103: Discuss this more, as it seems like these two statements are contradictory.

**Thank you, we have rewritten this part, now focusing on how to mitigate bias in the statistical learning methods. The (local) spatial groups are converged in sample size via this threshold adjustment with a slightly higher share of the urban group, aiming to compensate for the relatively high heterogeneity in this group.**

4. Line 135: 50000 estimators seems extremely excessive for the boosting algorithms. Traditionally, the number of estimators is in the hundreds. I would suggest the authors to discuss the reasoning behind this. While gradient boosting algorithms are resilient to overfit, they are not overfit-proof and including a very large number of estimators has the potential for overfitting.

**We appreciate the reviewer's concern regarding the use of 50,000 estimators in the boosting algorithms. While this number may appear excessive compared to traditional practices, it is important to note that boosting algorithms, such as gradient boosting, are designed to iteratively optimize residuals at each step. Boosting inherently uses the residuals from all observations to build the next tree, and if the gradient no longer descends (i.e., it reaches a minimum), the predictions stabilize, preventing further changes regardless of the number of estimators. This behavior is a fundamental difference from very large individual decision trees, which can overfit by continually splitting into smaller segments using local optimizers. Thus, while increasing the number of estimators significantly may extend computation time, it does not inherently increase the risk of overfitting. Instead, it provides the model with the flexibility to fully converge on the training data (Vezhnevets & Barinova, 2007). See also:**

https://datascience.stackexchange.com/questions/11272/is-boosting-resistant-to-overfitting-for-both-number-of-iterations-and-number-of

https://tomatofox.wordpress.com/2021/01/30/why-a-very-large-number-of-trees-wont-overfit-boosting/

**Vezhnevets, A., & Barinova, O. (2007). Avoiding boosting overfitting by removing confusing samples. In *Machine Learning: ECML 2007: 18th European Conference on Machine Learning, Warsaw, Poland, September 17-21, 2007. Proceedings 18* (pp. 430-441). Springer Berlin Heidelberg.**

5. Line 138, 141: Specific references to the supplementary material is missing. e.g. In which table are these results presented?

**We now added "section parameters".**

6. Line 150: SI table 4 shows the results for Linear, LASSO and Ridge and SI Table 5 shows the results for Random Forest, LightGBM etc.

**Thank you for this observation. We changed table numbers in supplementary and references in main accordingly.**

7. Section 2.2.2 and 2.2.3: The description of the methodology for the modeling part is superficial. The authors should expand these sections significantly, especially section 2.2.3 which describes the mixed-effects model and the Kriging method.

**Thank you for your feedback regarding the need for further elaboration in Sections 2.2.2 and 2.2.3. We agree that more detail on the methodology for the mixed-effects model and kriging would be beneficial to clarify the approach. We don't want the main to extent with too much information to guard simplicity and compactness – therefore we mainly extended only section 2.2.3.**

**In Section 2.2.2 on Multiple Linear Regression, we utilize regularization techniques such as LASSO and Ridge regression to prevent overfitting. These methods enable the selection of the most important predictors, while optimizing for model performance through tuning.**

**For Section 2.2.3 concerning the Mixed-Effects Model and Kriging, we have provided a more detailed explanation of the role of each method in our analysis**

8. Section 2.3: The SHAP values plot should be in the main text of the manuscript instead of the supplementary material, as it contains useful information that are used in the main study.

**Added in main. Every fold is visible in the supplementary material (Figure 9).**

9. Section 2.3: It's not clear whether the authors have normalised the data used here for the machine learning algorithms. From the SHAP figure it seems that the target variable range is not normalized. Normalization of the input and target variables to the 0 to 1 range ensures that all the input variables are equally weighted (unless the setup of the model specifically requires asymmetric weights) and the input variables with the largest range (or absolute values) do not dominate the others.

**For tree-based models like RandomForestRegressor, normalization is typically not required for shaply value calculation because they do not rely on distance metrics and coefficients of a linear regression model. This also becomes evident as one of the most important features, identified through the 10-fold repeated random sampling validation, contains very small values (the feature "trop mean filt"), see also table 2 in**

main. See also:

**Mohammadi, A., Karimzadeh, S., Banimahd, S. A., Ozsarac, V., & Lourenço, P. B. (2023). The potential of region-specific machine-learning-based ground motion models: application to Turkey. Soil Dynamics and Earthquake Engineering, 172, 108008.**

**Junda, E., Málaga-Chuquitaype, C., & Chawgien, K. (2023). Interpretable machine learning models for the estimation of seismic drifts in CLT buildings. Journal of Building Engineering, 70, 106365.**

**And following complementary article:**

https://shap.readthedocs.io/en/latest/example_notebooks/overviews/An%20introduction%20to%20explainable%20AI%20with%20Shapley%20values.html

10. Section 2.3: Consider expanding this section as it's not clear which predictors are selected and how the process is implemented. Also, a table of all the predictors used and their origin, range and name could be useful to the reader.

**The process of feature selection is guided by the out-of-sample performance in a 10-fold repeated random sampling validation, where Shapley values are calculated in each iteration of the random forest models. The median of every feature is used to determine the order from least influential to most influential feature on NO2 predictions. The specific features are now mentioned in section 2.3. Moreover, a table of global (2) and local (3) predictors (with descriptive statistics) are apparent in main.**

11. Line 192: In Kerckhoffs et. al (2019) the maps were produced by measuring for limited time periods using mobile sensors. The temporal resolution of the predictions of the models here deal with much coarser temporal resolutions.

**Thank you for your comment. We acknowledge that the temporal resolution of this benchmark data differs from the coarser temporal scales used in our models. The Kerckhoffs et al. (2019) data represents measurements over specific, limited time periods, while our models address predictions over broader temporal spans. Despite this temporal inconsistency, the detailed spatial granularity of the Kerckhoffs et al. map provides valuable insights and remains an appropriate standard for assessing spatial prediction quality. We added this acknowledgement in the revised version (section 2.4).**

**Results**

1. Line 201: How was this 20-fold validation performed? Since the number of data points is small, I suggest to perform cross-validation with a small number of folds (e.g. 5). How many data points were selected in each fold and on which set these metrics were evaluated on?

**Twenty-fold validation was performed by repeatedly splitting the dataset into training (75%) and testing (25%) subsets, using 20 different random states to ensure diverse splits (we now mention this division in section 2.4 and 3.1). Each fold maintained the same test size to ensure consistency, and performance metrics ($R^2$, MAE, RMSE) were evaluated on the test sets for each model. This approach provided robust estimates of model performance while accounting for variability due to data splitting.**

2. A graph of ground truth vs predictions would be beneficial here to identify edge cases at which the algorithms do not perform well.

**To provide additional insights, we have added supplementary 26 and 27.**

3. Line 216: It's not clear what the spatial resolution of the predictions is nor how these maps were created.

**Thank you for this observation, we can indeed clarify the resolution of the prediction area by mentioning it in the methodology (before, it was only apparent in the figure description). The resolution is 100m. TIF files are converted to 100m grid cells for the different regions of Amsterdam, Bayreuth, Hamburg, and Utrecht. Since we have the most influential predictor information (for global models, see table 2 and 4 on which predictors this are; for local models, table 3 and 4) available at 100m for the extent of above mentioned regions, we can use this information to predict the NO2 values for the respective 100m grids, based on the trained local and global models. We added this elaboration in the main (section 2.4).**

4. Table 6: Another useful metric that could be used to gauge the significance of the RMSE and MAE metrics would be percentage error wrt to ground truth.

**See supplementary 26 and 27.**

5. Line 256: When was leave-one-out cross-validation used before this? Discuss how this was implemented.

**Leave-one-out cross validation is opted because of the limited number of observations. Applying the 75/25 train/test methodology is valid for the global dataset due to a larger number of observations but the local dataset contains to few samples to obtain stable testing results. We have added this now in the methodology of main (section 2.4).**

6. Line 260: A significant limitation of the study setup is the fact that the most heterogeneous group (urban) is the least represented in terms of number of data points. This should be discussed by the authors.

**That is a good point. This is particularly a limitation for the global dataset. We updated the discussion of the main and acknowledge this by stating that an imbalance between relatively few samples and high heterogeneity cause poor performance. The urban group is more adequately represented in the local dataset**

7. The large number of models and the use of different set of models for each group makes these comparisons very difficult. Also, why have the authors used similar models (e.g. LightGBM and XGBoost) which makes the comparisons even more difficult to follow.

**You make a good point that this number of models makes the comparison harder to follow. Therefore, we removed the LightGBM analysis from main and only added this in the supplementary with references to it in main.**

8. Line 290: "These outliers were removed..." Discuss this choice more. How many points were removed and why do you think these points performed poorly?

The poor performance of these points could be attributed to several factors. One possible reason is out-of-range predictor values; regions with unusual predictor combinations may not be well-represented in the training data, leading to unreliable predictions. Below is a copy of the summary. The count of NO2 values below zero and 85 or above (i.e. classified as outliers) are shown, as well as the number of "not a number"-values. The results per model are visible.

**Omission Summary Details**

**Total number of samples: 10746**

**Variable-specific omission counts:**

**Model: random forest (global)**

 **- Not Meeting Criteria (<0 or >=85): 0**

 **- NAs: 0**

**Model: XGBoost (global)**

 **- Not Meeting Criteria (<0 or >=85): 0**

 **- NAs: 0**

**Model: LASSO (global)**

 **- Not Meeting Criteria (<0 or >=85): 0**

 **- NAs: 0**

**Model: Ridge (global)**

 **- Not Meeting Criteria (<0 or >=85): 0**

 **- NAs: 0**

**Variable: Linear (local)**

 **- Not Meeting Criteria (<0 or >=85): 0**

 **- NAs: 0**

**Model: Linear separating for spatial groups (local)**

 **- Not Meeting Criteria (<0 or >=85): 30**

 **- NAs: 0**

**Model: Mixed-effects model (local)**

 **- Not Meeting Criteria (<0 or >=85): 0**

 **- NAs: 0**

**Model: universal kriging (local)**

 **- Not Meeting Criteria (<0 or >=85): 0**

 **- NAs: 0**

**Model: universal kriging separating for spatial groups (local)**

**- Not Meeting Criteria (<0 or >=85): 30**

**- NAs: 0**

**Model: ordinary kriging (local)**

**- Not Meeting Criteria (<0 or >=85): 0**

**- NAs: 66**

**Discussion**

1. Line 323 "While a well-trained model..." Not clear what this sentence conveys to the reader.

**In the discussion of main, we have changed it to: "argue that the growing popularity of global models, due to their ability to capture both linear and non-linear relationships, may lead to misinformation. Although global models can make accurate predictions in regions where the predictor variables are well-represented in the training data, their performance may degrade in areas with predictor values that deviate significantly from the training range, highlighting the risk of spatial bias in predictions."**

2. LIne 334: I would argue that spatial cross-validation is essential in this kind of models, as it ensures that the model learns sufficient representations to generalise to other regions that do not have any ground stations. In this case, spatial cross-validation would be beneficial within the groups selected. i.e. ensure that the model generalises well within the urban cluster

**Thank you for this comment. We agree with your comment that the cross-validation methods is highly relevant for assessing the predictive ability of models. We are aware of the literature and critiques regarding spatial cross validation, as well as various cross validation methods. However, we considered spatial cross validation methods not suitable for our study. The reason is well explained in two discussions regarding spatial cross validation methods, Wadoux (2021) and Lu (2023). We agree with the arguments in these two papers consider randomly bootstrapped cross validation suitable to the accuracy assessment of our study.**

3. Line 342-346: This is counter-intuitive. It pollutes the urban group by including areas with less population that the initial definition (upper 75% quartile) but it was necessary to expand the size of the dataset to a sufficient level. The authors should make this clear and address it as a limitation of the study. While it's understandable this was a necessary step in the experimental setup of the study, it needs to be clearly addressed.

**In main, changed to:**

**"In the local dataset, the threshold for defining "urban" areas was adjusted from the upper 75% quartile (0.75) to the median (0.5). This adjustment was necessary due to the limited sample size, which required a broader definition to ensure sufficient data coverage for urban areas. However, this change also resulted in a less stringent definition of "urban," potentially including areas with lower population densities. While this adjustment expands the number of training samples available for the most heterogeneous group (urban), it introduces a limitation by diluting the urban group**

**and affecting the comparability of results. This trade-off underscores the challenges of balancing data representation with statistical robustness in spatial analyses."**

**Conclusions**

1. "In this study, we understand..." Consider changing the word understand to investigate

**We have changed this to "investigate" as recommended.**

**Review 2**

Thank you for the comprehensive revisions. This version effectively highlights varying performances across different spatial groups, which may inspire a critical and interesting reconsideration of current validation methodologies in air pollution modeling. I particularly like the methods section, as it provides details on the hyperparameters tuning. I have no major comments, only a few minor suggestions.

- Line 14:
The term "overfitting" is not correct. Please consider replacing it.

**We have changed this term. Now the relevant section states: "The spatial prediction patterns of global models show that non-linear methods generally are less sensitive to extreme values compared to linear methods."**

- Lines 16-17:
If I understand correctly, this work did not build models specific to spatial groups. Could you clarify the phrase, "but modeling spatial groups does," as its reference is unclear?

**We have clarified the phrase, changing it to "Using the local dataset of our study, explicitly accounting for spatial autocorrelation in the universal and ordinary kriging models does not improve accuracy; however, analyzing prediction performance across spatial groups provides valuable insights."**

- Line 329:
It is unclear why the statement suggests that nonlinear models outperform linear models in rural areas due to the homogeneous distribution of air pollution levels in these regions. My understanding is that linear models are typically more suitable for fitting homogeneous distributions.

**It might be more suitable when there is no enough data, as they sufficiently fit the data and provide clear interpretation. But when data is sufficient and the covariates are expressive to the response, the nonlinear model may provide an equivalent or even better fit if the relationship deviate from linear.**

- Lines 331-332:
The reasoning provided here seems unsupported. I do not see how this result substantiates the statement made.

**We have removed that part.**

- Lines 347-350:
These lines lack informativeness and could be deleted.

**Thanks for your observation on this, we removed that part.**

- Line 385:
The sentence here appears disconnected from the preceding context. Consider deleting it or adding a transition, like "Moreover."

**Removed.**

---

## Author Response (AR4)

Thank you for the latest revision. The manuscript has progressed well, but there are still a few issues that need to be addressed. In addition to the reviewer's comments, please consider the following:

- The captions of the new Tables 2 and 3 are certainly helpful in understanding Table 1, but please clarify already in Table 1 that the table provides statistics on station characteristics and that the distances in the "Variable" column represent buffer radii.

**We have included your suggestion in the table title.**

- Please include a simple snapshot of the GitHub repository in the zenodo archives (the clear structure and data files of the repository are not in the archives).

**There does not seem to be an easy way to add snapshots via images in the zenodo archives. There is a direct link to the repository on GitHub under the section "Additional details". As an alternative, I added the snapshot in the supplementary material zip.**

- https://doi.org/10.5281/zenodo.7948161 seems to have been superseded by https://zenodo.org/records/8219003.

**We updated the link to the zenodo archives: https://zenodo.org/records/15008748**

- Please exclude the cover letter from the supplementary zip file, which will be published as is.

**Review 1**

I would like to thank the authors for the revised manuscript and for their detailed replies to the comments they received. The revised manuscript addresses most of my previous concerns. There are however a couple of points that I believe were not addressed adequately before publication:

1. While it's true that gradient boosting algorithms optimize residuals iteratively, having too many estimators can lead to overfitting even after apparent convergence. The argument that "predictions stabilize" ignores that small changes in later trees can still accumulate to overfit the training data and it's the reason that XGBoost includes regularization terms, as unlimited (or very large number of) boosting rounds can lead to overfitting. The authors should justify the inclusion of such a high number of estimators and provide evidence (i.e. Validation curves). The reasoning that further trees that necessary don't affect the model is not valid, as later trees can still make small adjustments that collectively overfit. The authors can provide some empirical justification for this choice, like a curve of model performance with models trained on an increasing number of estimators until 50000. This can demonstrate whether 50,000 estimators were indeed necessary for convergence or not. The cited paper (Vezhnevets & Barinova, 2007) doesn't directly address modern gradient boosting implementation best practices.

**We have investigated the number of estimators further via a validation curve, as suggested:**

[Figure]

**With a number of 10,000 estimators, the performance of the xgboost model seems to stagnate. Having 50,000 estimators is therefore unnecessary. In the new version, the xgboost model contains 10,000 estimators.**

2. The 20-fold cross-validation methodology followed does not adhere to the standard k-fold cross validation methodology, where the data is divided into k non-overlapping folds, where each data point appears exactly once in the test set. The methodology used in the manuscript adheres more to the Monte-Carlo Cross-Validation method and not k-fold cross-validation. The random split employed can lead to biased performance estimates, especially with small datasets (some points may be used multiple times while others might not be used at all). I recommend to use fewer folds (5-fold cross validation is generally a good balance between bias and variance and make sure each point is used once.

**You are right that our explanation can be adjusted to fit the situation better. We abstain from using "fold" and prefer words such as "times" and "bootstrapping" to describe the situation. We still adhere to the bootstrapped CV approach (rather than using the x-fold divide). The bootstrapped CV aims at reduce the bias because of the random drawing of training points. We have reduced the number of testing samples for evaluating global model performances, meaning that the train/test split is now 90/10 instead of 75/25. Moreover, we chose to use bootstrapped CV because:**

**- it provides a robust measure of uncertainty for relatively small datasets.**

**- it works well when data is limited, as it generates multiple datasets through resampling.**

**In the discussion, we now also discuss the importance of choosing different CV approaches can make a difference in the results.**

3. The manuscript would benefit from clearer methodology descriptions, particularly regarding the mixed-effects and kriging models. The authors should also provide more detailed information about data resolution and how predictions were generated at 100m resolution. The limitation of having the most heterogeneous group (urban) being the least represented in terms of data points should be more thoroughly discussed.

Since the previous iteration, we elaborated more on the theories of mixed-effects and kriging models (section 2.2.3). We have expanded the discussion of both the mixed-effects and kriging models to provide more detailed explanations of the theoretical underpinnings. Specifically, we have clarified the role of fixed and random effects in the mixed-effects models and how they are used to account for spatial variation. In the case of kriging, we have elaborated on both ordinary and universal kriging methods used for local modeling, including how they account for spatial dependencies. At the same time, we explain the conversion of predictor information to a 100m by 100m grid so that we can make predictions via local and global models for the same resolution. Although we did our best to make an objective representation of the land use categories and for comparison between methods, the lack of observations of urban areas may be a cause of the unsatisfactory model performance. Note that the final prediction maps are unaffected by how we separate different land categories and can provide us additional perspectives in model performance, as well as the degree of details that we could expect. Also, we acknowledge the limitations of having the most heterogeneous group (urban) being the least represented in terms of global data points and/or represented by few local data points in the methodology (119-124), results (306-309), and discussion(390-405) sections.

We also agree that we may elaborate further on the points mentioned by you, but at the same time we have to keep our paper compact where possible as the research already has several extensive analyses.

---

## Author Response (AR5)

Thank you for the latest revision. The manuscript has progressed well, but there are still a few issues that need to be addressed. In addition to the reviewer's comments, please consider the following:

- The captions of the new Tables 2 and 3 are certainly helpful in understanding Table 1, but please clarify already in Table 1 that the table provides statistics on station characteristics and that the distances in the "Variable" column represent buffer radii.

**We have included your suggestion in the table title.**

- https://doi.org/10.5281/zenodo.7948161 seems to have been superseded by https://zenodo.org/records/8219003.

There was a misunderstanding about the Git repository snapshot: please include a zip archive of the Git repository (e.g. from "Download ZIP" on GitHub) that represents the version used for the manuscript in the Zenodo repository (and then update the Zenodo link in the Code and Data Availability section accordingly).

**Thank you for your clarification. We updated the link to the zenodo archives regarding the main scripts and data <100MB: (https://doi.org/10.5281/zenodo.15193954) and data repository for files >100MB (https://doi.org/10.5281/zenodo.15194548)**

Regarding the new paragraph on the Influence of Cross-Validation Techniques:

Page 26, line 396: "90/10 train-test split" should more clearly read "random 90/10 train-test split"

**Changed according to your suggestion.**

Line 403ff: The term "bootstrapping" often implies sampling with replacement, but apparently here the data were randomly split 90/10, which means sampling without replacement. Therefore, the term Monte Carlo cross-validation used by the reviewer seems more appropriate, and I suggest that the terminology be adjusted accordingly.

**Changed according to your suggestion.**

**Review 1**

I would like to thank the authors for the revised manuscript and for their detailed replies to the comments they received. The revised manuscript addresses most of my previous concerns. There are however a couple of points that I believe were not addressed adequately before publication:

1. While it's true that gradient boosting algorithms optimize residuals iteratively, having too many estimators can lead to overfitting even after apparent convergence. The argument that "predictions stabilize" ignores that small changes in later trees can still accumulate to overfit the training data and it's the reason that XGBoost includes regularization terms, as unlimited (or very large number of) boosting rounds can lead to overfitting. The authors should justify the inclusion of such a high number of estimators and provide evidence (i.e. Validation curves). The reasoning that further trees that necessary don't affect the model is not valid, as later trees can still make small adjustments that collectively overfit. The authors can provide some empirical justification for this choice, like a curve of model performance with models trained on an increasing number of estimators until 50000. This can demonstrate whether 50,000 estimators were indeed necessary for convergence or not. The cited paper (Vezhnevets & Barinova, 2007) doesn't directly address modern gradient boosting implementation best practices.

**We have investigated the number of estimators further via a validation curve, as suggested:**

[Figure]

**With a number of 10,000 estimators, the performance of the xgboost model seems to stagnate. Having 50,000 estimators is therefore unnecessary. In the new version, the xgboost model contains 10,000 estimators.**

2. The 20-fold cross-validation methodology followed does not adhere to the standard k-fold cross validation methodology, where the data is divided into k non-overlapping folds, where each data point appears exactly once in the test set. The methodology used in the manuscript adheres more to the Monte-Carlo Cross-Validation method and not k-fold cross-validation. The random split employed can lead to biased performance estimates, especially with small datasets (some points may be used multiple times while others might not be used at all). I recommend to use fewer folds (5-fold cross validation is generally a good balance between bias and variance and make sure each point is used once.

**You are right that our explanation can be adjusted to fit the situation better. We abstain from using "fold" and prefer words such as "times" and "iterations" to describe the situation. We changed the naming to Monte Carlo Cross-Validation (CV) and stick to this methodology, rather than using the x-fold divide. The Monte Carlo CV aims at reducing the bias because of the random drawing of training points. We have reduced the number of testing samples for evaluating global model performances, meaning that the random train/test split is now 90/10 instead of 75/25. Moreover, we chose to use Monte Carlo CV because:**

**- it provides a robust measure of uncertainty for relatively small datasets.**

**- it works well when data is limited, as it generates multiple datasets through resampling.**

**In the discussion, we now also discuss the importance of choosing different CV approaches can make a difference in the results.**

3. The manuscript would benefit from clearer methodology descriptions, particularly regarding the mixed-effects and kriging models. The authors should also provide more detailed information about data resolution and how predictions were generated at 100m resolution. The limitation of having the most heterogeneous group (urban) being the least represented in terms of data points should be more thoroughly discussed.

**Since the previous iteration, we elaborated more on the theories of mixed-effects and kriging models (section 2.2.3). We have expanded the discussion of both the mixed-effects and kriging**

models to provide more detailed explanations of the theoretical underpinnings. Specifically, we have clarified the role of fixed and random effects in the mixed-effects models and how they are used to account for spatial variation. In the case of kriging, we have elaborated on both ordinary and universal kriging methods used for local modeling, including how they account for spatial dependencies. At the same time, we explain the conversion of predictor information to a 100m by 100m grid so that we can make predictions via local and global models for the same resolution. Although we did our best to make an objective representation of the land use categories and for comparison between methods, the lack of observations of urban areas may be a cause of the unsatisfactory model performance. Note that the final prediction maps are unaffected by how we separate different land categories and can provide us additional perspectives in model performance, as well as the degree of details that we could expect. Also, we acknowledge the limitations of having the most heterogeneous group (urban) being the least represented in terms of global data points and/or represented by few local data points in the methodology (119-124), results (306-309), and discussion(390-405) sections.

We also agree that we may elaborate further on the points mentioned by you, but at the same time we have to keep our paper compact where possible as the research already has several extensive analyses.